# On the Hardness of Conditional Independence Testing In Practice

**Zheng He**
UBC
zhhe@cs.ubc.ca

**Roman Pogodin**
McGill and Mila
rmn.pogodin@gmail.com

**Yazhe Li**
Microsoft AI*
yazheli@outlook.com

**Namrata Deka**
CMU
ndeka@andrew.cmu.edu

**Arthur Gretton**
Gatsby Unit, UCL
arthur.gretton@gmail.com

**Danica J. Sutherland**
UBC and Amii
dsuth@cs.ubc.ca

## Abstract

Tests of conditional independence (CI) underpin a number of important problems in machine learning and statistics, from causal discovery to evaluation of predictor fairness and out-of-distribution robustness. Shah and Peters (2020) showed that, contrary to the unconditional case, no universally finite-sample valid test can ever achieve nontrivial power. While informative, this result (based on "hiding" dependence) does not seem to explain the frequent practical failures observed with popular CI tests. We investigate the Kernel-based Conditional Independence (KCI) test – of which we show the Generalized Covariance Measure underlying many recent tests is *nearly* a special case – and identify the major factors underlying its practical behavior. We highlight the key role of errors in the conditional mean embedding estimate for the Type-I error, while pointing out the importance of selecting an appropriate conditioning kernel (not recognized in previous work) as being necessary for good test power but also tending to inflate Type-I error.

## 1 Introduction

Conditional independence (CI) testing is a fundamental task, required for almost any scientific hypothesis that "controls for" confounders; it is moreover a core subroutine in the standard PC algorithm for causal discovery and its many variants (Spirtes et al., 1993). Further recent major machine learning-specific applications include checking or enforcing the fairness of a predictor or representation with equalized odds (Hardt et al., 2016), and relatedly for a predictor's domain invariance, particularly in "anticausal" settings (e.g. Lu et al., 2021).

When the conditioning variable takes on a small number of discrete values, the problem is simple to reduce to that of unconditional independence testing, for which there are many good methods: for instance, many based on the Hilbert-Schmidt Independence Criterion (HSIC) (Gretton et al., 2005; Gretton et al., 2007). When the conditioning variable is continuous, however, the situation is much more challenging: when testing whether $A \perp\!\!\!\perp B \mid C$ based on samples for a continuously distributed $C$,[1] we will only observe one $(A, B)$ pair for each value of $C$, and so we must make some form of assumption on the smoothness of the conditional distribution $(A, B) \mid C = c$ as a function of $c$. Shah and Peters (2020) proved that doing so in total generality is impossible. Their lower bound, however, is an adversarial construction of a particular distribution (discussed in Section 4) which does not seem especially informative as to the widespread failures of CI tests in practical settings. Since the importance of the task means that, despite its impossibility in general, we still want to pursue CI testing, we must consider particular types of tests used in practice and when, and why, they fail.

---

*Work done at Gatsby Unit, UCL.

[1]We will always use $A \perp\!\!\!\perp B \mid C$, since papers in this area use both $X \perp\!\!\!\perp Y \mid Z$ and $X \perp\!\!\!\perp Z \mid Y$.

39th Conference on Neural Information Processing Systems (NeurIPS 2025).

There are a few major categories of techniques. One is the Kernel-based Conditional Independence (KCI) technique introduced by K. Zhang et al. (2011). As a kernel method, this technique is applicable to data of any (potentially complex and structured) form. It has a reputation, however, of doing a poor job at controlling Type-I error: that is, it falsely identifies conditional dependence too often (Shah and Peters, 2020; Pogodin et al., 2024). Recent extensions include CIRCE (Pogodin et al., 2023), which is useful as a regularizer for learning $A$ but generally yields a much worse test, and SplitKCI (Pogodin et al., 2024), which helps reduce Type-I error rates, but is far from solving the issue. Related approaches include Scetbon et al. (2022), who characterize conditional independence using analytic kernel embeddings evaluated at finitely many locations, and Y. Zhang et al. (2025), who study conditional mean independence—essentially the KCI statistic with a linear kernel on A.

A number of studies propose to test conditional independence by checking the covariance of residuals from regressions of $A$ and $B$ on $C$ (e.g., H. Zhang et al., 2017; H. Zhang et al., 2018; Shah and Peters, 2020). We refer to this class of methods collectively as the Generalized Covariance Measure (GCM), following Shah and Peters (2020). While conceptually simple, GCM captures only linear covariance between residuals and averages the dependence over $C$, rather than evaluating the covariance conditional on specific values of $C$. Weighted GCM (Scheidegger et al., 2022) generalizes the GCM by applying weights based on $C$, allowing detection of a broader range of conditional dependencies. As we show in Section 3, the standard GCM corresponds to a special case of KCI with simple kernel choices, while Weighted GCM can be viewed as a more flexible, though still constrained, setting of the $C$ kernel.

Having introduced measures of conditional independence, we revisit some theoretical work on the CI testing hardness in Section 4, where in particular we show that challenges in CI testing with kernel statistics arise specifically due to challenges in estimating the *conditional mean embedding*, a kernel embedding of the conditional distribution that underpins the majority of such tests (Song et al., 2009; Grünewälder et al., 2012; Klebanov et al., 2020; Park and Muandet, 2020; Li et al., 2024). In Section 5, we provide a clear demonstration that choosing an appropriate $C$ kernel is vital to a sensitive KCI test – in contrast to an implicit claim by K. Zhang et al. (2011) and the approach taken by Pogodin et al. (2023) and Pogodin et al. (2024). Following related work in other settings (e.g. Jitkrittum et al., 2016; Liu et al., 2020; Xu et al., 2024), we suggest a method to select a $C$ kernel which does help achieve more powerful tests. We observe, however, that this method can also make the problem of false rejection even more severe.

In Section 6, we investigate the problem of false rejections in KCI tests. We first analyze simple yet informative special cases, which allow analytical investigation of how regression errors in estimating conditional mean embeddings induce bias in the test statistic's moments. These insights motivate a more general theoretical analysis, where we derive formal bounds linking conditional mean estimation error to test validity. Together, the results clarify the root cause of false rejections and delineate the conditions under which KCI and GCM tests remain reliable.

## 2   Measuring Conditional Dependence

We first show how to measure conditional dependence with kernels. While the fundamental idea is due to K. Zhang et al. (2011), our framing is somewhat different in terms of the novel Theorem 2.2.

**Conditional independence.**   We build on the characterization of Daudin (1980). To begin, we formalize the intuition that given $C$, $A$ and $B$ contain no additional information about one another:

**Definition 2.1** (Daudin, 1980)**.**  Random variables $A$ and $B$ are conditionally independent given $C$, denoted $A \perp\!\!\!\perp B \mid C$, if for all square-integrable functions $f \in L^2_{AC}$ and $g \in L^2_{BC}$,

$$\mathbb{E}[\, f(A,C)\, g(B,C) \mid C\,] = \mathbb{E}[\, f(A,C)|C] \; \mathbb{E}[\, g(B,C) \mid C\,] \quad \text{almost surely in } C.$$

This definition is equivalent to stating that the conditional joint distribution factorizes almost surely in $C$, $P_{A,B|C} = P_{A|C} P_{B|C}$, by considering functions $f$ and $g$ as indicators of events.

Building on this definition, we can derive the following equivalence for conditional independence:

**Theorem 2.2.** *Random variables $A$ and $B$ are conditionally independent given $C$ if and only if*

$$\mathbb{E}_C \left[ w(C) \; \mathbb{E}_{AB|C} \big[ \, (f(A) - \mathbb{E}[f(A) \mid C]) \, (g(B) - \mathbb{E}[g(B) \mid C]) \mid C \big] \right] = 0, \tag{1}$$

*for all square-integrable functions $f \in L^2_A$, $g \in L^2_B$, and $w \in L^2_C$.*

This result, proved in Appendix A, extends the characterization of Daudin (1980) to a particularly interpretable form: does any residual dependence between $A$ and $B$ remain after accounting for $C$? The weighting function $w(C)$ allows emphasizing specific regions of the support of $C$. Under $A \perp\!\!\!\perp B \mid C$, the conditional covariances vanish $C$-almost surely; otherwise, there is some nonzero conditional covariance on a $C$-non-negligible region, which an appropriate $w(C)$ can capture.

**Kernel spaces.** Since it is infeasible to check all square-integrable functions for $f$, $g$, and $w$, we instead focus on a restricted yet sufficiently rich class of "smooth" functions. Specifically, we consider functions that lie in reproducing kernel Hilbert spaces (RKHSs), which enable characterization of conditional dependence via kernel mappings.

A reproducing kernel Hilbert space (RKHS) $\mathcal{H}_{\mathcal{A}}$ is a particular space of functions $\mathcal{A} \to \mathbb{R}$; each RKHS is uniquely associated to a positive-definite kernel $k_A : \mathcal{A} \times \mathcal{A} \to \mathbb{R}$. This kernel can itself be written as $k_A(a, a') = \langle \phi_A(a), \phi_A(a') \rangle_{\mathcal{H}_{\mathcal{A}}}$, where $\phi_A : \mathcal{A} \to \mathcal{H}_{\mathcal{A}}$ is known as a feature map. The defining *reproducing property* of an RKHS is that for all $f \in \mathcal{H}_{\mathcal{A}}$ and $a \in \mathcal{A}$, $f(a) = \langle f, \phi_A(a) \rangle_{\mathcal{H}_{\mathcal{A}}}$. We always assume that any RKHS we deal with is separable; this is guaranteed when $k$ is continuous and the underlying space $\mathcal{A}$ is separable (Steinwart and Christmann, 2008, Lemma 4.33).

**KCI operator.** The following operator, introduced (in a different form) by K. Zhang et al. (2011), will help us characterize conditional dependence; we reframe it, following Theorem 2.2, to explicitly incorporate a conditional covariance structure.[2] We build this up in pieces.

First, the conditional mean embeddings $\mu_{A|C}(c) := \mathbb{E}[\phi_A(A) \mid C = c] \in \mathcal{H}_{\mathcal{A}}$ and $\mu_{B|C}(c) := \mathbb{E}[\phi_B(B) \mid C = c] \in \mathcal{H}_{\mathcal{B}}$ provide RKHS representations of the conditional distributions of $A$ and $B$ given $C = c$. They satisfy the reproducing property $\langle \mu_{A|C}(c), f \rangle_{\mathcal{H}_{\mathcal{A}}} = \mathbb{E}[f(A) \mid C = c]$.

The conditional cross-covariance operator, $\mathfrak{C}_{AB|C}$, will capture the dependence structure between $A$ and $B$ with $\langle f, \mathfrak{C}_{AB|C}(c)g \rangle = \mathbb{E}_{AB|C}\left[ \left( f(A) - \mathbb{E}[f(A) \mid C] \right) \left( g(B) - \mathbb{E}[g(B) \mid C] \right) \mid C = c \right]$:

$$\mathfrak{C}_{AB|C}(c) := \mathbb{E}_{AB|C}\left[ \left( \phi_A(A) - \mu_{A|C}(c) \right) \otimes \left( \phi_B(B) - \mu_{B|C}(c) \right) \mid C = c \right] \in \mathrm{HS}(\mathcal{H}_{\mathcal{B}}, \mathcal{H}_{\mathcal{A}}). \quad (2)$$

Here $\mathrm{HS}(\mathcal{H}_{\mathcal{B}}, \mathcal{H}_{\mathcal{A}})$ denotes the space of Hilbert–Schmidt operators from $\mathcal{H}_{\mathcal{B}}$ to $\mathcal{H}_{\mathcal{A}}$, and the outer product $\phi_A(a) \otimes \phi_B(b) \in \mathrm{HS}(\mathcal{H}_{\mathcal{B}}, \mathcal{H}_{\mathcal{A}})$ is defined by $\left( \phi_A(a) \otimes \phi_B(b) \right)g = \langle \phi_B(b), g \rangle_{\mathcal{H}_{\mathcal{B}}} \phi_A(a)$ for any $g \in \mathcal{H}_{\mathcal{B}}$, analogous to the outer product of vectors in finite-dimensional spaces.

The KCI operator aggregates these conditional covariances with information about the context $C$:

$$\mathfrak{C}_{\mathrm{KCI}} := \mathbb{E}_C\left[ \mathfrak{C}_{AB|C}(C) \otimes \phi_C(C) \right] \in \mathrm{HS}(\mathcal{H}_{\mathcal{C}}, \mathrm{HS}(\mathcal{H}_{\mathcal{B}}, \mathcal{H}_{\mathcal{A}})). \quad (3)$$

For any test functions $f \in \mathcal{H}_{\mathcal{A}}$, $g \in \mathcal{H}_{\mathcal{B}}$ and $w \in \mathcal{H}_{\mathcal{C}}$, the properties above give that

$$\langle f \otimes g, \mathfrak{C}_{\mathrm{KCI}}\, w \rangle_{\mathrm{HS}(\mathcal{H}_{\mathcal{B}}, \mathcal{H}_{\mathcal{A}})} = \mathbb{E}_C\left[ w(C) \, \mathbb{E}_{AB|C}\left[ \left( f(A) - \mathbb{E}[f(A) \mid C] \right) \left( g(B) - \mathbb{E}[g(B) \mid C] \right) \right] \right].$$

If the KCI operator is itself zero, then the quantity above is zero for any choice of $f \in \mathcal{H}_{\mathcal{A}}$, $g \in \mathcal{H}_{\mathcal{B}}$, $w \in \mathcal{H}_{\mathcal{C}}$. If the KCI operator is nonzero, then there exist $f, g, w$ for which it is nonzero, implying that $A \not\perp\!\!\!\perp B \mid C$. A natural measure of conditional dependence is then the magnitude of $\mathfrak{C}_{\mathrm{KCI}}$, as measured by its squared Hilbert-Schmidt norm:

$$\mathrm{KCI} := \|\mathfrak{C}_{\mathrm{KCI}}\|^2_{\mathrm{HS}} = \mathbb{E}_{C,C'}\left[ k_C(C, C') \left\langle \mathfrak{C}_{AB|C}(C), \mathfrak{C}_{AB|C}(C') \right\rangle_{\mathrm{HS}(\mathcal{H}_{\mathcal{B}}, \mathcal{H}_{\mathcal{A}})} \right]. \quad (4)$$

The Hilbert–Schmidt norm of an operator is zero if and only if the operator itself is the zero operator. If the RKHSs $\mathcal{H}_{\mathcal{A}}$, $\mathcal{H}_{\mathcal{B}}$ and $\mathcal{H}_{\mathcal{C}}$ are $L^2$-universal, meaning that they are dense in $L^2$, then $\mathrm{KCI} = 0$ if and only if $A \perp\!\!\!\perp B \mid C$. Many standard choices, such as the Gaussian RBF kernel $k_A(a, a') = \exp(-\|a - a'\|^2/(2\ell^2))$, are $L^2$-universal (cf. Sriperumbudur et al., 2011; Szabó and Sriperumbudur, 2018), where $\ell$ is the kernel lengthscale. Large values of KCI indicate strong evidence of conditional dependence, while values near zero suggest that any apparent dependence can be adequately explained by $C$.

---

[2]To obtain this formulation from theirs: first, following Pogodin et al. (2023), remove the $C$ to $C$ regression of the original version (also see Mastouri et al., 2021, Appendix B.9). Second, use a product kernel on $(B, C)$; we are not aware of any uses that do *not* do this, and our framing of Theorem 2.2 makes the final product clearer.

# 3 Connecting KCI and GCM

Shah and Peters (2020) proposed a Generalized Covariance Measure, which has been the basis of many recent CI tests (Hochsprung et al., 2023; Wieck-Sosa et al., 2025). For scalar $A$ and $B$, GCM uses a studentized estimate of the average covariance between residuals, based on any regression method from $C$ to $A$. Scheidegger et al. (2022) extend the approach to Weighted GCM, which adds a weighting function $w$; assuming perfect regressions, the population quantity becomes

$$\mathbb{E}\big[\, w(C)(A - \mathbb{E}[\, A \mid C\,])\, (B - \mathbb{E}[\, B \mid C\,])\,\big] \tag{5}$$

With $w(c) = 1$, this is the quantity estimated by GCM; an appropriate choice of $w$ function increases the sensitivity to more types of dependence.

Consider KCI with scalar linear kernels $\phi_A(a) = a$ and $\phi_B(b) = b$. This makes the conditional mean embeddings $\mu_{A|C}(c) = \mathbb{E}[\phi_A(A) \mid C = c] = \mathbb{E}[A \mid C = c]$, and similarly $\mu_{B|C}(c) = \mathbb{E}[B \mid C = c]$. If we further pick the kernel $k_C(c, c') = w(c)w(c')$ so $\phi_C(c) = w(c)$, then (3) becomes identical to (5). The difference is that GCM estimates the value of that expectation (normalized by the standard deviation of the estimates), while the KCI operator estimates the absolute value. This relationship is analogous to that between classifier two-sample tests and maximum mean discrepancy-based tests (Liu et al., 2020, Section 4), and to that between variational mutual information-based independence tests and HSIC tests (Xu et al., 2024).

Consider instead $\mathcal{A} = \mathbb{R}^{d_A}$, $\mathcal{B} = \mathbb{R}^{d_B}$, with multivariate linear[3] $\phi_A(a) = a$, $\phi_B(b) = b$ and the same $\phi_C = w$. The conditional cross-covariance (2) becomes the conditional cross-covariance matrix of shape $d_A \times d_B$, and the KCI operator (3) is the $w$-weighted average of that matrix. The multivariate (weighted) GCM again takes a studentized estimate of that matrix, and uses the maximum absolute value as its entry. KCI would instead use the Frobenius norm.

In this way, we can see that (weighted) GCM is almost a special case of KCI using simple kernels, further motivating our study of KCI in particular (especially with linear $\phi_A$ and $\phi_B$). The advantage of the weighted over the unweighted statistic also foreshadows the importance of $k_C(c, c')$.

# 4 Revisiting the Theoretical Hardness of CI Testing

In null hypothesis significance testing, we seek a test that rejects the null, i.e. claims that $A \not\perp\!\!\!\perp B \mid C$ (the alternative), with no more than $\alpha$ probability (say 0.05) when the null hypothesis that $A \perp\!\!\!\perp B \mid C$ is in fact true. Such rejections, also known as false positives, are called *Type-I errors*. A test has (finite-sample) valid level if its Type-I error rate is at most $\alpha$, while it has (pointwise) asymptotically valid level if for any null distribution, the Type-I error rate is asymptotically no more than $\alpha$. Failing to reject the null when it does not hold is called a *Type-II error*; the *power* of a test is the rate at which it does reject, i.e. one minus the Type-II error rate. Among valid tests, the best one is the one with the highest power. A test is *consistent against fixed alternatives* if for any distribution where the null does not hold, the power approaches 1 as $n \to \infty$.

**Impossibility result.** Shah and Peters (2020) showed that if a CI test has finite-sample valid level for all Lebesgue-continuous null distributions, then it has power no more than $\alpha$ for any Lebesgue-continuous alternative. This is in stark contrast to the unconditional case (or conditioning on a discrete variable), in which case there exist finite-sample valid, consistent tests (e.g. permutations based on HSIC; see Rindt et al., 2021).

Intuitively, when detecting unconditional dependence $A \perp\!\!\!\perp B$, dependence can be missed (causing a Type-II error) but Type-I error arises only from sampling variability. By contrast, for $A \perp\!\!\!\perp B \mid C$, it is possible either to miss actual dependence (Type-II) or falsely detect dependence (Type-I) because subtle conditional effects of C have been overlooked. For the latter case, consider generating $C, A', B' \sim \mathcal{N}(0, 1)$, extracting the thirtieth decimal place of $C$ as $C_{30} \in \{0, 1, \dots, 9\}$, and then taking $A = C_{30} + A'$, $B = C_{30} + B'$. Unless we know to look at the thirtieth decimal place of $C$, $A$ and $B$ will seem to be strongly dependent and $C$ irrelevant; in fact, however, all information that $A$ carries about $B$ is present in $C$, so $A \perp\!\!\!\perp B \mid C$. Shah and Peters (2020) show that for all

---

[3]The formulation as in (2) should have $\phi_A(a) \in \mathcal{H}_A$, i.e. the function $a' \mapsto \langle a, a' \rangle$; here and in the following paragraph we identify $\mathbb{R}^d$ with its dual by instead using $a$, which yields the same KCI value and other quantities.

test procedures, for any case which is truly conditionally dependent, the test has such a "blind spot" which is conditionally independent but "looks the same" to the test.

**Interpretation with KCI.** How do these issues manifest with KCI? We can show, in fact, that they arise *solely* because of the estimation of the conditional mean embedding.

In practice, conditional independence testing relies on empirical estimates constructed from finite samples. Given observations $\{(a_i, b_i, c_i)\}_{i=1}^n$, we first define the KCI statistic $\mathrm{KCI}_n$ as a U-statistic based on the true conditional mean embeddings $\mu_{A|C}$ and $\mu_{B|C}$:

$$\mathrm{KCI}_n = \frac{1}{n(n-1)} \sum_{1 \leq i \neq j \leq n} h_{i,j} \qquad \text{where } h_{i,j} = (K_C)_{i,j} \, (K_A^{\mathrm{c}})_{i,j} \, (K_B^{\mathrm{c}})_{i,j}, \tag{6}$$

where $(K_C)_{i,j} = k_C(c_i, c_j)$ is the kernel matrix for $C$, $(K_A^{\mathrm{c}})_{i,j} = \langle \phi_A^{\mathrm{c}}(a_i, c_i), \phi_A^{\mathrm{c}}(a_j, c_j) \rangle_{\mathcal{H}_A}$ with $\phi_A^{\mathrm{c}}(a_i, c_i) = \phi_A(a_i) - \mu_{A|C}(c_i)$ is the centered kernel matrix for $A$, and similarly $(K_B^{\mathrm{c}})_{i,j} = \langle \phi_B^{\mathrm{c}}(b_i, c_i), \phi_B^{\mathrm{c}}(b_j, c_j) \rangle_{\mathcal{H}_B}$ with $\phi_B^{\mathrm{c}}(b_i, c_i) = \phi_B(b_i) - \mu_{B|C}(c_i)$ is that for $B$.

To run a KCI-based test, we require a *test threshold* $t_n$ and reject the null whenever the KCI statistic exceeds $t_n$. This threshold $t_n$ is selected based on an estimate of the null distribution of the statistic, which depends on the sample size $n$, the choice of kernels, and the underlying data distribution. K. Zhang et al. (2011) show that when $A \perp\!\!\!\perp B \mid C$, $n\mathrm{KCI}_n$ converges to a mixture of $\chi^2$ variables,[4] so $t_n$ could in principle be estimated by fitting the parameters of this limiting distribution. If we know the true $\mu_{A|C}$ and $\mu_{B|C}$, we can easily construct a finite-sample valid test with nontrivial power:

**Proposition 4.1.** *Suppose* $\sup_{a \in \mathcal{A}} k_A(a, a) \leq \kappa_A$, $\sup_{b \in \mathcal{B}} k_B(b, b) \leq \kappa_B$, $\sup_{c \in \mathcal{C}} k_C(c, c) \leq \kappa_C$. *Then a test which rejects when* $\mathrm{KCI}_n > \tilde{t}_n \coloneqq 32\kappa_A\kappa_B\kappa_C\sqrt{\frac{1}{n-1}\log\frac{1}{\alpha}}$ *has finite-sample level at most $\alpha$. Moreover, if each kernel is $L^2$-universal, the test is consistent against fixed alternatives.*

The proof, given in Appendix B, is a simple consequence of Hoeffding's inequality for $U$-statistics. Although the resulting test is highly conservative – the correct threshold for the null distribution of $\mathrm{KCI}_n$ should be $\Theta(1/n)$ (K. Zhang et al., 2011, Theorem 3), much smaller than the chosen $\tilde{t}_n$ – the fact that it avoids the impossibility result of Shah and Peters (2020) indicates that the main challenge lies in estimating conditional mean embeddings.

**Relationship to model-X.** The recently popular "model-X" setting (Candes et al., 2018; Berrett et al., 2019; Grünwald et al., 2024) assumes that the conditional distribution of $A \mid C$ is known. This corresponds to perfect knowledge of $\mu_{A|C}$: for a characteristic (or a fortiori, $L^2$-universal) $k_A$, $\mu_{A|C}$ uniquely corresponds to $\mathrm{Law}(A \mid C)$. Given knowledge of both $A \mid C$ and $B \mid C$, the KCI-based test in Proposition 4.1 would be exactly valid; knowledge of only one is also sufficient using CIRCE rather than KCI (Pogodin et al., 2023; Pogodin et al., 2024). We discuss more aspects of the relationship to other CI tests in Appendix C.

## 5 Pitfalls of Kernel Choices for CI Testing in Practice

Since the true conditional mean embeddings are unknown, in practice we must use the empirical KCI statistic $\widehat{\mathrm{KCI}}_n$, which substitutes these embeddings with estimates $\widehat{\mu}_{A|C}$ and $\widehat{\mu}_{B|C}$. These embeddings are typically estimated via kernel ridge regression (Grünewälder et al., 2012; Li et al., 2024) with inputs $c_i$ and labels $\phi_A(a_i)$ or $\phi_B(b_i)$. KCI requires a choice of as many as five kernels to operate. The original formulation (K. Zhang et al., 2011) used the same kernel for both regression steps, but Pogodin et al. (2023) noted that high-quality regressions typically demand different kernels. They therefore proposed choosing separate kernels via leave-one-out validation, introducing two additional regression kernels, denoted $k_{C \to A}$ and $k_{C \to B}$. Pogodin et al. (2023) and Pogodin et al. (2024) then used $k_C$ as either $k_{C \to A}$ or $k_{C \to B}$, implicitly assuming that a good kernel for this regression will also be a good kernel for measuring dependence.

---

[4]Their Proposition 5 makes a stronger claim, that $\widehat{\mathrm{KCI}}_n$ does so under fixed-regularization ridge regression estimates for the conditional means; their argument (which was only sketched) appears to rely on a property that does not clearly always hold for this estimator, but does hold with the true $\mu_{A|C}, \mu_{B|C}$. Personal communication with the authors confirmed that they agree "there is a gap" between the published sketch and a true proof.

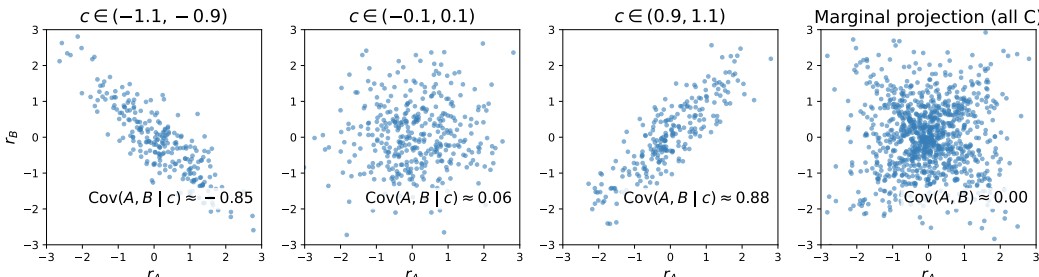

Figure 1: **Motivating example.** We simulate $(r_A, r_B, C)$ following problem (7), where $\tau = 1$ and the residual correlation $\gamma(C) = \sin(C)$ introduces dependence that varies smoothly with $C$. The left three panels visualize the residuals for different slices of $C$, showing that $\mathrm{Cov}(A, B \mid C)$ changes substantially across $C$. The rightmost panel shows the residuals for all values of $C$, where the averaged conditional covariance $\mathbb{E}_C[\mathrm{Cov}(A, B \mid C)]$ is zero. A kernel on $C$ with an appropriate lengthscale can focus on local regions where dependence is strong: too long a lengthscale "blurs" the conditional covariance, while too short a lengthscale leaves too little data to estimate it reliably.

We now demonstrate that the aforementioned choice for $k_C$ – though computationally convenient – can be a very poor choice for measuring dependence in complex situations. For an intuitive example, consider an engineering problem involving high-dimensional vibration data: we wish to know if the behavior of part $A$ is connected to that of part $B$ given vibration data $C$. While predicting the behavior of either $A$ or $B$ depends on broad, long-term trends of $C$, the two parts may be coupled only by high-frequency sinusoidal resonances which require a substantially different kernel to efficiently detect. Using $k_{C \to A}$ or $k_{C \to B}$ then results in high Type-II error.

Motivated by this, consider a synthetic problem where $A$ and $B$ are determined as some functions of $C$ plus noise factors which are zero mean, but potentially conditionally correlated given $C$:

$$C \sim \mathcal{N}(0,1), \quad A = f_A(C) + \tau\, r_A, \quad B = f_B(C) + \tau\, r_B,$$

where $f_A, f_B$ are fixed functions, $\tau, \beta > 0$, and the additive residual terms $(r_A, r_B)$ follow

$$(r_A, r_B) \mid C \sim \mathcal{N}\left( \begin{bmatrix} 0 \\ 0 \end{bmatrix}, \begin{bmatrix} 1 & \gamma(C) \\ \gamma(C) & 1 \end{bmatrix} \right), \quad \gamma(C) = \begin{cases} 0 & \text{under } \mathfrak{H}_0 \\ \sin(\beta C) & \text{under } \mathfrak{H}_1. \end{cases} \tag{7}$$

We use linear kernels for $A$ and $B$, aligning closely to GCM, and a lengthscale-$\ell_C$ Gaussian kernel $k_C(C, C') = \exp\left( -\frac{(C - C')^2}{2\ell_C^2} \right)$ on $C$; GCM corresponds to $\ell_C = \infty$.

Figure 1 illustrates this setup under $\mathfrak{H}_1$: although the conditional covariance $\mathrm{Cov}(A, B \mid C) = \tau^2\, \mathbb{E}[r_A r_B \mid C]$ changes smoothly with $C$ and alternates in sign, its expectation $\mathbb{E}_C[\mathrm{Cov}(A, B \mid C)]$ is zero. Methods based on this average, like GCM, thus fail to detect dependence, highlighting the need for kernels on $C$ that can localize to regions where the conditional covariance is nonzero.

The regressions estimating $f_A$ and $f_B$ should use kernels $k_{C \to A}$ and $k_{C \to B}$ with lengthscales suited to those functions. In contrast, the kernel $k_C$ used for the residuals should target the lengthscale of the covariance function $\gamma$ (i.e., $1/\beta$), which can differ substantially from the regression lengthscales.

In this setting, we can analytically evaluate the KCI, at least when using the true mean embeddings $\mu_{A|C}$ and $\mu_{B|C}$. Details are given in Appendix F.1. We first see, using properties of Gaussians, that

$$\mathrm{KCI} = \tau^4 \mathop{\mathbb{E}}_{C,C'} \left[ k_C(C, C')\gamma(C)\gamma(C') \right] = \tau^4 \sqrt{\frac{\ell_C^2}{\ell_C^2 + 2}} \mathop{\mathbb{E}}_{(X,X') \sim \mathcal{N}_{\ell_C}} \left[ \gamma(X)\gamma(X') \right] \tag{8}$$

for auxiliary variables $(X, X') \sim \mathcal{N}_{\ell_C} := \mathcal{N}\left( \begin{bmatrix} 0 \\ 0 \end{bmatrix}, \begin{bmatrix} 1 - \frac{1}{\ell_C^2 + 2} & \frac{1}{\ell_C^2 + 2} \\ \frac{1}{\ell_C^2 + 2} & 1 - \frac{1}{\ell_C^2 + 2} \end{bmatrix} \right)$. Under the null, we of course obtain $\mathrm{KCI} = 0$; under the alternative, we can use trigonometric identities to see

$$\mathrm{KCI} = \frac{1}{2}\tau^4 e^{-\beta^2} \sqrt{\frac{\ell_C^2}{\ell_C^2 + 2}} \left( e^{2\beta^2/(\ell_C^2 + 2)} - 1 \right).$$

When $\ell_C \ll \sqrt{2}$, the square root term arising from $k_C(C, C')$ vanishes, giving zero KCI; for $\ell_C \gg \beta$, the other term coming from the covariance of $\gamma$ vanishes, yielding the same problem. Consequently,

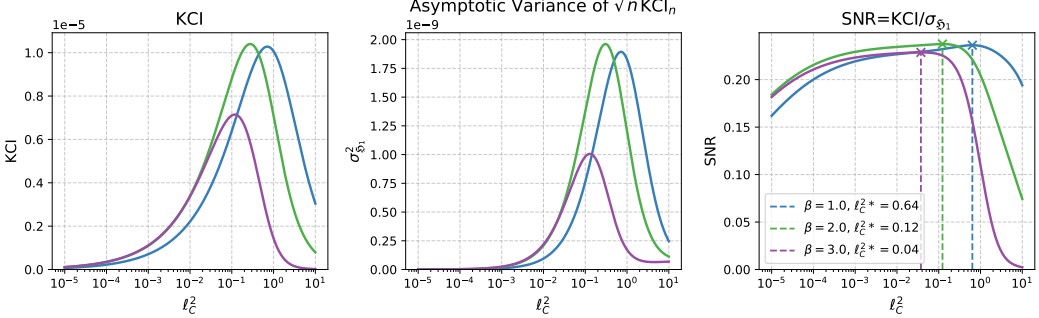

Figure 2: Effect of squared kernel lengthscale $\ell_C^2$ on KCI, asymptotic variance $\sigma_{\mathfrak{H}_1}^2$, and approximate test power (SNR) for different conditional covariance frequency $\beta$, in the synthetic example (7) under the alternative. The optimal $\ell_C^2{}^*$ is selected by maximizing SNR. Different $\beta$ values correspond to different $\ell_C^2$ ranges yielding high approximate test power. Here we use noise scale $\tau = 0.1$.

for each $\beta$, the effective $\ell_C$ lies at an intermediate value that balances these effects (see Figure 2, left). GCM, with $\ell_C = \infty$, cannot detect dependence here at all.

**Selecting a conditioning kernel.** How can we choose the right $C$ kernel for a given problem? One approach, following that taken in related settings (Jitkrittum et al., 2016; Liu et al., 2020; Xu et al., 2024), is to maximize the approximate power of the test, based on the following asymptotic result:

**Proposition 5.1.** *Under the alternative, there is a scalar $\hat{\sigma}_{\mathfrak{H}_1}^2 \geq 0$ so that as $n \to \infty$,*

$$\sqrt{n}(\widehat{\mathrm{KCI}}_n - \widehat{\mathrm{KCI}}) \xrightarrow{d} \mathcal{N}(0, \hat{\sigma}_{\mathfrak{H}_1}^2). \tag{9}$$

As always, the hat here refers to the use of estimated mean embeddings, not to estimation of a quantity from samples; $\widehat{\mathrm{KCI}}$ and $\hat{\sigma}_{\mathfrak{H}_1}^2$ depend on the problem, the kernels, and the choice of $\widehat{\mu}_{A|C}$ and $\widehat{\mu}_{B|C}$, but not on $n$ or any particular test sample. Under the alternative, we typically have $\hat{\sigma}_{\mathfrak{H}_1}^2 > 0$, in which case the rejection probability is approximately

$$\mathrm{Pr}_{\mathfrak{H}_1}(\widehat{\mathrm{KCI}}_n > t_n) \sim \Phi\left( \frac{\sqrt{n}\,\widehat{\mathrm{KCI}}}{\hat{\sigma}_{\mathfrak{H}_1}} - \frac{\sqrt{n}\,t_n}{\hat{\sigma}_{\mathfrak{H}_1}} \right),$$

where $a \sim b$ means $\lim_{n\to\infty} a/b = 1$, $\Phi$ is the standard normal CDF, and $t_n$ is any rejection threshold. We expect $t_n = \Theta(1/n)$, following the null distribution of $\mathrm{KCI}_n$; the power is therefore dominated by the first term for reasonably large $n$, and the kernel yielding the most powerful test will approximately maximize the signal-to-noise ratio $\widehat{\mathrm{SNR}} = \widehat{\mathrm{KCI}}/\hat{\sigma}_{\mathfrak{H}_1}$.

We estimate $\widehat{\mathrm{SNR}}$ as the ratio of $\widehat{\mathrm{KCI}}_n$ to its estimated standard deviation (Liu et al., 2020, Eq. (5)), and choose the kernel on a training split that maximizes this value. (In independent work, Wang et al. (2025) used a similar scheme, but with a somewhat different estimator setup and with limited analysis; see Appendix C.1.) We can then use the selected kernel on a testing split; as long as the two splits are independent, this will not break the independence assumptions of the test procedure.

For a fixed $\widehat{\mu}_{A|C}$ and $\widehat{\mu}_{B|C}$, $\widehat{\mathrm{SNR}}_n$ in fact generalizes, identifying a good kernel:

**Theorem 5.2** (Informal). *Consider the $U$-statistic kernel $\hat{h}$ of $\widehat{\mathrm{KCI}}_n$; give it parameters $\omega$, such as the parameters of $k_C$, in a finite-dimensional Banach space such that $\hat{h}$ is smooth with respect to those parameters. Then $\widehat{\mathrm{SNR}}_n$ converges uniformly to $\widehat{\mathrm{SNR}}$ over bounded sets of parameters with variance bounded away from zero; thus the maximizer of $\widehat{\mathrm{SNR}}_n$ approaches that of $\widehat{\mathrm{SNR}}$.*

This is a modification of the result of Liu et al. (2020, Theorem 6), since for fixed $\widehat{\mu}_{A|C}, \widehat{\mu}_{B|C}$ the $U$-statistic structure is very similar; a detailed statement and a proof are in Appendix D.

To evaluate whether maximizing the approximate test power is effective in practice, we compare the theoretical (approximate) power with the empirical power estimated from data. Figure 2 illustrates how the analytic results KCI, $\sigma_{\mathfrak{H}_1}^2$, and the corresponding SNR vary with the squared kernel

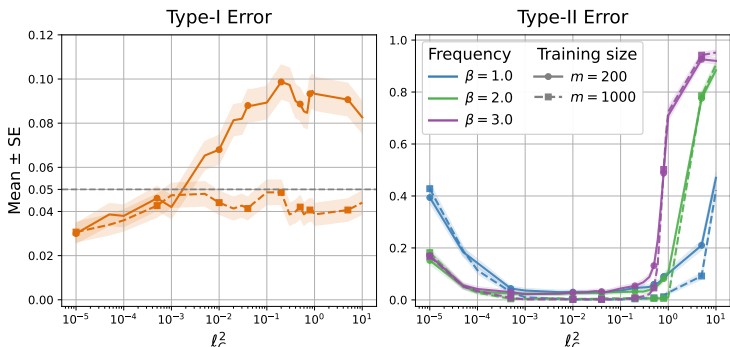

Figure 3: Means and standard error (over 500 runs) of Type-I/II errors on the synthetic example (7) with $f_A = \cos, f_B = \exp, \tau = 0.1$ and different values of $\beta$, plotted against the squared kernel lengthscale $\ell_C^2$. The training sample size is $m = 200$ (solid line with circles) or $m = 1000$ (dashed line with squares); the independent test set has size 200. The significance level is set at $\alpha = 0.05$. Details about testing procedure is in Appendix H.1. **Left**: When $m = 200$, varying $\ell_C^2$ noticeably affects the Type-I error, for certain values of $\ell_C^2$. In contrast, when $m = 1000$, the regressor is better trained, and the Type-I error remains well-controlled for all $\ell_C^2$. **Right**: The empirical test power depends strongly on both $\beta$ and $\ell_C^2$, indicating the the importance of proper kernel selection for $k_C$.

lengthscale $\ell_C^2$ in the synthetic example (7), where the optimal $\ell_C^{2^*}$ is obtained by maximizing the SNR. As shown in Figure 3 (right), the theoretical power curve (SNR vs. $\ell_C^2$) closely tracks the empirical power curve ($(1 - \text{Type-II error})$ vs. $\ell_C^2$), indicating that the selected $\ell_C^{2^*}$ remains effective in practice under the alternative hypothesis.

Although maximizing test power is effective under the alternative, it can substantially inflate Type-I error in CI testing. In the unconditional settings of Liu et al. (2020) and Xu et al. (2024), the null threshold is determined by permutation, ensuring exact Type-I error control: any chosen kernel rejects at most at rate $\alpha$. In our case, no such procedure is available; instead, we rely on asymptotic null approximations, which depend sensitively on kernel choice and regression quality, making null calibration delicate. Train/test splitting prevents overfitting to the points used to select $\ell_C^2$, yet $\ell_C^2$ can still overfit to the imperfect regressors $\widehat{\mu}_{A|C}$ and $\widehat{\mu}_{B|C}$. As shown in Figure 3 (left), Type-I error remains controlled when these regressors are well trained; with limited training data, however, there exists a range of $\ell_C^2$ values where it rises sharply. Power maximization tends to favor this region due to its higher $\widehat{\text{SNR}}$. The $\ell_C^2$ is then selected to capture the spurious dependence caused by regression errors rather than the true signal. Hence, when the conditioning kernel is chosen based on noisy regressions, an inherent tension arises between Type-I error control and test power.

**Relationship to weighted GCM selection scheme.** One approach of Scheidegger et al. (2022) identifies a weight function by predicting the sign of residuals' product. If that prediction works perfectly, then it changes GCM from measuring the average residual correlation to measuring its average absolute value, which is potentially much more powerful. As discussed, this is essentially equivalent to selecting $k_C$, which they set as a $\pm 1$ indicator based on whether the residuals have the same predicted sign. While the scheme works differently than ours, it has essentially the same trade-offs as other approaches for kernel selection.

## 6 Type-I Error Inflation with Regression Errors

As shown by Proposition 4.1 and reinforced by the previous section's example, the fundamental challenges in conditional independence testing stem from the estimation of conditional mean embeddings. To further illustrate this point, we examine the effect of regression errors by letting $\widehat{\mu}_{A|C} = \mu_{A|C} + \Delta_{A|C}$ and $\widehat{\mu}_{B|C} = \mu_{B|C} + \Delta_{B|C}$. Under the null hypothesis, we can explicitly characterize $\widehat{\text{KCI}}$ and its asymptotic variance in terms of $\Delta_{A|C}$ and $\Delta_{B|C}$. This enables us to quantify how regression errors distort the KCI statistic and, consequently, to establish formal bounds that relate estimation error to test validity.

**Effect on moments.** The following result, proved in Appendix E, is a more convenient form of textbook results about $U$-statistics (Serfling, 1980, Section 5.2.1) for kernel methods:

**Theorem 6.1.** *Let* $h(X, X') = \langle \phi_h(X), \phi_h(X') \rangle$ *with mean embedding* $\mu_h = \mathbb{E}_X \phi_h(X)$ *and the centered covariance operator* $\mathfrak{C}_h = \mathbb{E}_X [\phi_h(X) \otimes \phi_h(X)] - \mu_h \otimes \mu_h$. *Define* $\nu_1 = \langle \mu_h, \mathfrak{C}_h \mu_h \rangle$ *and* $\nu_2 = \|\mathfrak{C}_h\|_{\mathrm{HS}}^2$. *The corresponding* $U$-*statistic satisfies*

$$U_n = \frac{1}{n(n-1)} \sum_{1 \le i \ne j \le n} h(X_i, X_j), \quad \mathbb{E}[U_n] = U = \|\mu_h\|^2, \quad \mathrm{Var}(U_n) = \frac{4}{n} \nu_1 + \frac{2}{n(n-1)} \nu_2.$$

The function $h$ in (6), for $\mathrm{KCI}_n$, has this form with $\phi_h(X) = \phi_A^{\mathrm{c}}(A, C) \otimes \phi_B^{\mathrm{c}}(B, C) \otimes \phi_C(C)$, and its mean $\mu_h = \mathbb{E}_X[\phi_h(X)] = \mathbb{E}_C \left[ \mathbb{E}_{AB}[\phi_A^{\mathrm{c}}(A, C) \otimes \phi_B^{\mathrm{c}}(B, C) \mid C] \otimes \phi_C(C) \right]$ is exactly $\mathfrak{C}_{\mathrm{KCI}}$ in (3). Thus, under the null $\mu_h = \mathfrak{C}_{\mathrm{KCI}} = 0$, we have $\mathbb{E} \, \mathrm{KCI}_n = \mathrm{KCI} = 0$ and $\mathrm{Var}(\mathrm{KCI}_n) = \frac{2}{n(n-1)} \nu_2$.

$\widehat{\mathrm{KCI}}_n$ has the same decomposition, except $\mu_{\hat{h}} = \mathbb{E}_C \left[ \mathbb{E}_{AB}[\widehat{\phi}_A^{\mathrm{c}}(A, C) \otimes \widehat{\phi}_B^{\mathrm{c}}(B, C) \mid C] \otimes \phi_C(C) \right]$ is now *not* zero if the error in $\widehat{\mu}_{A|C}$, $\widehat{\mu}_{B|C}$ is not exactly conditionally independent. As shown by Pogodin et al. (2024), with linear kernels $k_A$ and $k_B$, under the null we have

$$\mathbb{E} \widehat{\mathrm{KCI}}_n = \widehat{\mathrm{KCI}} = \mathbb{E} \left[ k_C(C, C') \langle \Delta_{A|C}(C), \Delta_{A|C}(C') \rangle_{\mathcal{H}_A} \langle \Delta_{B|C}(C), \Delta_{B|C}(C') \rangle_{\mathcal{H}_B} \right]. \quad (10)$$

As they note, we typically expect $\Delta_{A|C}$ and $\Delta_{B|C}$ to be relatively smooth functions of $C$; thus it is reasonable to expect that $\widehat{\mathrm{KCI}}$ can be nontrivial even though they were trained on independent datasets. Perhaps even more significantly, for fixed regression functions, it will generally be the case that $\nu_1 = \langle \mu_{\hat{h}}, \mathfrak{C}_{\hat{h}} \mu_{\hat{h}} \rangle > 0$. This implies that the standard deviation decays only as $\Theta(1/\sqrt{n})$, rather than the faster $\Theta(1/n)$ obtained achieved $\Delta_{A|C}, \Delta_{B|C}$ are zero. The exact variance of $\widehat{\mathrm{KCI}}_n$ in terms of $\Delta_{A|C}$ and $\Delta_{B|C}$ is given in Appendix F.2. [5]

**Multi-dimensional $C$ example.** So far we implicitly presumed that the same features of $C$ are used both in the true/estimated conditional means and the dependence $\gamma(C)$. We thus extend our analysis to cases where $C$ is multi-dimensional, considering two scenarios: (1) using the same coordinates of $C$ for both conditional means and dependence, and (2) using separate coordinates. This allows us to study how the information in $C$ drives spurious dependence. See Appendix F.3 for the problem formulation and Appendix H.2 for the experimental setup and additional results.

Table 1: Comparison of Testing Results for Two Conditional Dependence Scenarios

| Scenario | Type-I Error | Type-II Error |
|---|---|---|
| Scenario 1: Shared coordinates | 0.21 | 0.0 |
| Scenario 2: Separate coordinates | 0.10 | 0.08 |

As observed in Table 1, Scenario 1 exhibits a notably higher Type-I error (0.21) compared to Scenario 2 (0.10). This increase arises from regression errors leaking correlated noise into the test statistic when regressions and dependence share the same coordinate. Consequently, Scenario 1 exhibits an increase in false positives. In contrast, Scenario 2, with independent dimensions, shows lower Type-I error but slightly higher Type-II error, illustrating a trade-off driven by correlated regression errors.

We further present real-world experiments in Appendix H.3. These observations motivate a closer look at how regression errors affect the theoretical behavior of KCI, particularly its null calibration.

**Effect on null calibration.** Standard methods for setting a test threshold for KCI do not incorporate regression error; rather, they rely on the asymptotic distribution of $\mathrm{KCI}_n$. For instance, K. Zhang et al. (2011) estimate the threshold using a $\chi^2$ mixture or a gamma approximation, while Pogodin et al. (2024) suggest a wild bootstrap approach. In either case, the null threshold scales as $\Theta(1/n)$. However, if regression errors remain fixed while the number of test points grows, $\widehat{\mathrm{KCI}}_n = \Theta(1) + \mathcal{O}_p(1/\sqrt{n})$ will almost surely exceed the threshold, inflating Type-I error. This shows that regression errors must shrink as $n$ grows, and motivates establishing the required decay rate.

---

[5]Although derived under simplifying assumptions on data distribution, it extends directly since the result is independent of choices of $f_A(C)$, $f_B(C)$, and Gaussianity of residuals.

**Asymptotics.** When $\nu_1 > 0$, $\sqrt{n}(U_n - U)$ converges to a normal distribution (Proposition 5.1); when $\nu_1 = 0$ but $\nu_2 > 0$, $n(U_n - U)$ converges in distribution to a weighted mixture of centered $\chi^2$ variables (Serfling, 1980, Section 5.5). We can thus ask: under the null, how likely is a sample from $\mathrm{KCI}_n$ to exceed a test threshold set according to the limiting distribution of $n\mathrm{KCI}_n$?

**Theorem 6.2.** *Assume that $A \perp\!\!\!\perp B \mid C$. Let $Z_1 = \widehat{\mathrm{KCI}}_n$ and $Z_2 \sim \mathcal{N}\left(\widehat{\mathrm{KCI}}, \mathrm{Var}(\widehat{\mathrm{KCI}}_n)\right)$ be a normal variable moment-matched to $Z_1$. Let $q > 0$ and $\rho \in (0, 1)$; define $T_1 = \sqrt{(1-\rho)/\rho}$ and $T_2 = \Phi^{-1}(1 - \rho)$, where $\Phi$ is the standard normal CDF. Then the following holds for $i \in \{1, 2\}$:*

$$\Pr\left(Z_i > \frac{q}{n}\right) \le \rho \quad \text{if } q \ge n\widehat{\mathrm{KCI}} + T_i \sqrt{n^2 \, \mathrm{Var}(\widehat{\mathrm{KCI}}_n)}.$$

The proof is in Appendix G.1; the case for $\widehat{\mathrm{KCI}}_n$ is more precisely applicable, but using asymptotic normality gives better dependence on $\rho$. This theorem provides an upper bound on the probability that the inflated statistic $\widehat{\mathrm{KCI}}_n$ exceeds a nominal null threshold. Intuitively, the bound shows that if the regression bias induced $\widehat{\mathrm{KCI}}$ or the variance $\mathrm{Var}(\widehat{\mathrm{KCI}}_n)$ are non-negligible, the effective threshold $q/n$ must grow proportionally to $\widehat{\mathrm{KCI}} + T_i\sqrt{\mathrm{Var}(\widehat{\mathrm{KCI}}_n)}$ in order to maintain the level $\rho$.

In practice, we use the wild bootstrap to approximate the null distribution of the KCI statistic. Formally, it generates surrogate samples $Y = \frac{1}{n}\sum_{i \ne j} \hat{h}_{ij}\, \varepsilon_i \varepsilon_j$, where $\varepsilon_i$ are independent noise[6] and $\hat{h}_{ij}$ is defined as in (6) using $\widehat{\mu}_{A|C}$ and $\widehat{\mu}_{B|C}$. For a given kernel matrix, $Y$ has zero mean and variance closely matching that of $n\mathrm{KCI}_n$ under the null. The following result, shown in Appendix G.2, bounds the approximation error between the wild bootstrap and a moment-matched normal for $n\widehat{\mathrm{KCI}}_n$.

**Theorem 6.3.** *Assume $A \perp\!\!\!\perp B \mid C$, and let $Y = \frac{1}{n}\sum_{i,j=1}^{n} \hat{h}_{ij}\, \varepsilon_i\, \varepsilon_j$, where $\varepsilon_i \overset{\mathrm{iid}}{\sim} \mathcal{N}(0, 1)$. Let $\widehat{H}$ be the matrix with entries $\hat{h}_{i,j}$; assume $\delta := \|\widehat{H}\|_\infty^2 / \|\widehat{H}\|_2^2 < 1/2$. Let $Z_n \sim \mathcal{N}\left(\widehat{\mathrm{KCI}}, \mathrm{Var}(\widehat{\mathrm{KCI}}_n)\right)$. Define the standardized mean shift $b_{\widehat{\mathrm{KCI}}} = \frac{\widehat{\mathrm{KCI}}}{\sqrt{\mathrm{Var}(\widehat{\mathrm{KCI}}_n)}}$ and variance mismatch $\kappa_{\mathrm{var}} = \frac{\mathrm{Var}(Y|\widehat{H})}{n^2\, \mathrm{Var}(\widehat{\mathrm{KCI}}_n)}$. Further define $R_1 = \frac{2\pi^2}{3}\mathrm{Skew}(Y \mid \widehat{H}) = \frac{4}{3}\sqrt{2}\,\pi^2\|\widehat{H}\|_3^3 / \|\widehat{H}\|_2^3$, $R_2 = \frac{2^{-5/4}\,\pi^{-2}}{\sqrt{1-2\,\delta}}$, and $R_3 = (2\pi)^{-7/2}$. Then, for $\Psi(x) = \frac{1}{\sqrt{2\pi}}\, x\, \exp(2\pi x)$,*

$$\sup_{x \in \mathbb{R}}\left|\Pr(Y \mid \widehat{H} \le x) - \Pr(n\, Z_n \le x)\right| \le \Psi\left(R_1 \kappa_{\mathrm{var}}^{3/2} + b_{\widehat{\mathrm{KCI}}} + \pi|\kappa_{\mathrm{var}} - 1|\right) + R_2 \kappa_{\mathrm{var}}^{-1/2} + R_3,$$

Noting $n^2\, \mathrm{Var}(\widehat{\mathrm{KCI}}_n) \sim 4n\nu_1 + 2\nu_2$, Theorem 6.3 is most meaningful if $\widehat{\mathrm{KCI}} = o(1/n)$, and $\nu_1 = o(1/n)$, so that the standardized mean shift $b_{\widehat{\mathrm{KCI}}} \to 0$ and the variance mismatch $\kappa_{\mathrm{var}} \to 1$. Consequently, Theorem 6.3 makes the asymptotic discrepancy between the wild-bootstrap statistic $Y$ and its Gaussian approximation $nZ_n$ converge as $n \to \infty$ to $\Psi(R_1) + R_2 + R_3$. This non-vanishing constant remains under perfect regression in part because of the mismatch between the normal $Z_n$ and the asymptotically mixture-of-chi-squareds $\mathrm{KCI}_n$, rather than (necessarily) errors in the wild bootstrap itself. Similarly, in Theorem 6.2, the asymptotic threshold behaves correctly when $4n\nu_1 + 2\nu_2 \to v \le (q/T_i)^2$, which is most easily achieved when $\nu_1 = o(1/n)$ and $\nu_2 = \Theta(1)$. Taken together, both bounds consistently require $\widehat{\mathrm{KCI}} = o(1/n), \nu_1 = o(1/n)$.

## 7  Discussion

We provided a novel framing of the KCI test, one which helped us connect it closely to GCM-based tests. We explained how this category of tests interacts with the famed hardness result of Shah and Peters (2020), identifying regression error as the key difficulty, and showing bounds on the excess Type-I error based on the amount of regression error. We showed that, contra the assumptions of most prior work, selecting a $k_C$ kernel specifically for testing can be of vital importance in achieving test power, but that doing so can exacerbate Type-I error.

While CI testing remains fundamentally difficult, our work makes a step towards understanding how this difficulty manifests in practice, and demonstrates paths towards addressing it. This underscores that users of GCM- or KCI-type tests must carefully consider how to mitigate spurious residual dependence under the null—something that sample splitting alone does *not* resolve.

---

[6]Pogodin et al. (2024) suggest using Rademacher $\varepsilon_i$; we use Gaussians in our analysis.

## Acknowledgments and Disclosure of Funding

The authors would like to thank Aaron Wei for productive discussions.

This work was supported in part by the Natural Sciences and Engineering Research Council of Canada, the Canada CIFAR AI Chairs program, the Gatsby Charitable Foundation, Calcul Québec, the BC DRI Group, the Digital Research Alliance of Canada, and a Google research gift.

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

## A  Conditional Independence Decomposition

We recall and prove Theorem 2.2, which extends results of Daudin (1980).

**Theorem 2.2.** *Random variables $A$ and $B$ are conditionally independent given $C$ if and only if*

$$\mathbb{E}_C\left[w(C)\ \mathbb{E}_{AB|C}\big[\,(f(A) - \mathbb{E}[f(A)\mid C])\,(g(B) - \mathbb{E}[g(B)\mid C])\mid C\big]\right] = 0, \qquad (11)$$

*for all square-integrable functions $f \in L_A^2$, $g \in L_B^2$, and $w \in L_C^2$.*

*Proof.* (i) Let $A$ and $B$ be conditionally independent given $C$. Let $\tilde{f} \in L_{AC}^2$ and $\tilde{g} \in L_{BC}^2$. Then by Definition 2.1, almost surely in $C$ it holds that

$$\mathbb{E}\big[\,\tilde{f}(A,C)\,\tilde{g}(B,C)\mid C\,\big] = \mathbb{E}\big[\,\tilde{f}(A,C)\mid C\,\big]\ \mathbb{E}\big[\,\tilde{g}(B,C)\mid C\,\big],$$

which is equivalent to the statement that almost surely in $C$,

$$\mathbb{E}_{AB|C}\big[\,(\tilde{f}(A,C) - \mathbb{E}[\,\tilde{f}(A,C)\mid C\,])\,(\tilde{g}(B,C) - \mathbb{E}[\,\tilde{g}(B,C)\mid C\,])\mid C\,\big] = 0.$$

Since this expectation is almost surely zero, it holds for any $w \in L_C^2$ that

$$\mathbb{E}_C\left[w(C)\ \mathbb{E}_{AB|C}\big[\,(\tilde{f}(A,C) - \mathbb{E}[\,\tilde{f}(A,C)\mid C\,])\,(\tilde{g}(B,C) - \mathbb{E}[\,\tilde{g}(B,C)\mid C\,])\mid C\,\big]\right] = 0.$$

Given any $f \in L_A^2$ and any $g \in L_B^2$, we can choose $\tilde{f}(\cdot, c) = f$ and $\tilde{g}(\cdot, c) = g$ to simply ignore the second argument. These functions satisfy $\tilde{f} \in L_{AC}^2$ and $\tilde{g} \in L_{BC}^2$. Then, as desired,

$$\mathbb{E}_C\left[w(C)\ \mathbb{E}_{AB|C}\big[\,(f(A) - \mathbb{E}[\,f(A)\mid C\,])\,(g(B) - \mathbb{E}[\,g(B)\mid C\,])\mid C\,\big]\right] = 0.$$

(ii) Suppose (11) holds for all functions $\tilde{f} \in L_A^2$, $\tilde{g} \in L_B^2$, and $\tilde{w} \in L_C^2$. Let $P_C$ denote the marginal distribution of $C$, and let $P_{A|C}$, $P_{B|C}$, and $P_{AB|C}$ denote the conditional distributions of $A$, $B$, and $(A, B)$ given $C$, respectively. Let $\mathcal{Q}$ be a Borel subset of the image set of $C$. Pick $w^* = \mathbb{1}_\mathcal{Q} \in L_C^2$, where $\mathbb{1}_\mathcal{Q}$ is the indicator function of $\mathcal{Q}$. Substituting this choice into equation (11) yields

$$\int_\mathcal{Q} \mathbb{E}_{AB|C}\big[\,(\tilde{f}(A) - \mathbb{E}[\,\tilde{f}(A)\mid C\,])\,(\tilde{g}(B) - \mathbb{E}[\,\tilde{g}(B)\mid C\,])\mid C\,\big]\,dP_C = 0,$$

Since this holds for all Borel sets $\mathcal{Q}$, it follows that the integrand must vanish almost surely with respect to $P_C$. That is, for $P_C$-almost every value of $C$, (11) implies that

$$\mathbb{E}\big[\,\tilde{f}(A)\,\tilde{g}(B)\mid C = c\,\big] = \mathbb{E}\big[\,\tilde{f}(A)\mid C = c\,\big]\ \mathbb{E}\big[\,\tilde{g}(B)\mid C = c\,\big].$$

Given any $f \in L_{AC}^2$ and any $g \in L_{BC}^2$, for each $C = c$ in its domain, $f(\cdot, c) \in L_A^2$ and $g(\cdot, c) \in L_B^2$ for almost every $c$. Thus, for any $f \in L_{AC}^2$ and any $g \in L_{BC}^2$, we have for almost every $c$,

$$\mathbb{E}\big[\,f(A,c)\,g(B,c)\mid C = c\,\big] = \mathbb{E}\big[\,f(A,c)\mid C = c\,\big]\ \mathbb{E}\big[\,g(B,c)\mid C = c\,\big],$$

which is precisely Definition 2.1. This completes the proof.  □

## B  Finite-sample Valid Test with Exact Mean Embeddings

We recall and prove Proposition 4.1.

**Proposition 4.1.** *Suppose $\sup_{a\in\mathcal{A}} k_A(a,a) \le \kappa_A$, $\sup_{b\in\mathcal{B}} k_B(b,b) \le \kappa_B$, $\sup_{c\in\mathcal{C}} k_C(c,c) \le \kappa_C$. Then a test which rejects when $\mathrm{KCI}_n > \tilde{t}_n := 32\kappa_A\kappa_B\kappa_C\sqrt{\frac{1}{n-1}\log\frac{1}{\alpha}}$ has finite-sample level at most $\alpha$. Moreover, if each kernel is $L^2$-universal, the test is consistent against fixed alternatives.*

*Proof.* $\text{KCI}_n$ is a $U$-statistic with kernel

$$k_C(c, c') \langle \phi_A(a) - \mu_{A|C}(c), \phi_A(a') - \mu_{A|C}(c') \rangle_{\mathcal{H}_A} \langle \phi_B(b) - \mu_{A|C}(c), \phi_A(b') - \mu_{A|C}(c') \rangle_{\mathcal{H}_B}.$$

We have that

$$\|\phi_A(a)\| = \sqrt{\langle \phi_A(a), \phi_A(a) \rangle} = \sqrt{k_A(a, a)} \le \sqrt{\kappa_A}$$

and by Jensen's inequality

$$\|\mu_{A|C}(c)\| = \big\| \mathbb{E}[\phi_A(A) \mid C = c] \big\| \le \mathbb{E}[\|\phi_A(A)\| \mid C = c] \le \sqrt{\kappa_A},$$

so that $\|\phi_A(a) - \mu_{A|C}(c)\| \le 2\sqrt{\kappa_A}$. Hence, by Cauchy-Schwarz,

$$|\langle \phi_A(a) - \mu_{A|C}(c), \phi_A(a') - \mu_{A|C}(c') \rangle_{\mathcal{H}_A}| \le 4\kappa_A.$$

Similarly, $|\langle \phi_B(b) - \mu_{B|C}(c), \phi_B(b') - \mu_{B|C}(c') \rangle_{\mathcal{H}_B}| \le 4\kappa_B$. Thus the kernel of the $U$-statistic $\text{KCI}_n$ has absolute value at most $16\kappa_A\kappa_B\kappa_C$. Hoeffding (1963)'s inequality for $U$-statistics (c.f. Serfling, 1980, Section 5.6.1, Theorem A) thus shows that when $\text{KCI} = 0$,

$$\Pr\left(\text{KCI}_n \ge t_n\right) \le \exp\left(-\frac{2\lfloor n/2 \rfloor t_n^2}{4 \cdot (16\kappa_A\kappa_B\kappa_C)^2}\right) \le \exp\left(-\frac{(n-1) t_n^2}{(32\kappa_A\kappa_B\kappa_C)^2}\right) = \alpha,$$

showing finite-sample validity of the test.

On the other hand, when $A \not\perp\!\!\!\perp B \mid C$, since each kernel is $L^2$-universal, we know that $\text{KCI} > 0$. Thus a symmetric application of Hoeffding's inequality tells us that once $n$ is large enough that $t_n < \text{KCI}/2$, we have that

$$\Pr\left(\text{KCI}_n < t_n\right) = \Pr\left(\text{KCI} - \text{KCI}_n > \text{KCI} - t_n\right)$$

$$\le \exp\left(-(n-1)\left(\frac{\text{KCI} - t_n}{32\kappa_A\kappa_B\sqrt{\kappa_C}}\right)^2\right) \le \exp\left(-(n-1)\left(\frac{\text{KCI}/2}{32\kappa_A\kappa_B\sqrt{\kappa_C}}\right)^2\right) \to 0,$$

and hence for any fixed alternative, the probability of a Type-II error goes to zero. $\square$

## C Relationship to Other Testing Methods

**Relationship to other CI tests.** One major category of conditional independence tests are based on variations of approximate permutation, i.e. that samples with similar $C$ values have similar $A$ and $B$ distributions, which can be exploited either by "swapping" samples with nearby $C$ values (e.g. Sen et al., 2017; Berrett et al., 2019; Kim et al., 2022) or by producing bins of $C$ values and assuming the distribution is constant within (e.g. Györfi and Walk, 2012). While this approach might seem fundamentally different than the regression or conditional mean embedding approaches, we emphasize that it is not. For instance, Kim et al. (2022) assume that the Hellinger or Rényi distance between $A \mid C = c$ and $A \mid C = c'$ is at most a constant times $\|c - c'\|$, and the same for $B$; similar assumptions underlie all methods of this type. This smoothness justifies using the distribution $A \mid C = c'$ to estimate $A \mid C = c$ for some similar value of $c'$. Bearing in mind the one-to-one correspondence between mean embeddings and distributions, this assumption is essentially equivalent to using a nearest-neighbor type estimator for $\widehat{\mu}_{A|C}, \widehat{\mu}_{B|C}$.

Another recent CI test is the Rao-Blackwellized Predictor Test, RBPT (Polo et al., 2023). This method is regression-based, but based on comparing predictors of $B \mid A, C$ to an averaged predictor of $B \mid C$. This structure makes it harder to compare to the KCI-type tests directly, but we note that it relies on a good estimate of $A \mid C$ and hence is essentially in the model-X framework. Like most tests in this area, it suffers from severe bias problems, as discussed by Pogodin et al. (2024).

**Smoothness of distributions.** In the model-X setting where the conditional distribution is only approximately known, Berrett et al. (2019, Section 5) bound the worst-case inflation of the Type-I error for two common model-X tests by at most the average conditional total variation distance between the true distribution and the approximation. Generic distribution modeling methods are likely to succeed in this sense only if the distribution changes slowly in total variation. Similarly, the bound of Kim et al. (2022) in a permutation case assumes that the distribution changes slowly in Hellinger distance; note that the total variation distance is upper-bounded by a constant times the

Hellinger distance, and so slow Hellinger change is a (slightly) stronger assumption than slow total variation change.

By contrast, while the precise conditions for effective conditional mean embedding estimation are complex (Li et al., 2024), we can roughly expect them to work when the mean embedding changes smoothly as a function of $C$; that is, the maximum mean discrepancy, MMD (Gretton et al., 2012), between $A \mid C = c$ and $A \mid C = c'$ changes slowly as a function of $c$. It is easy to see (e.g. Xu et al., 2024) that for bounded kernels, the MMD is a lower bound on the total variation. Thus, in the settings where the bounds of Berrett et al. (2019) and Kim et al. (2022) are applicable, we can *roughly* expect that $\mu_{A|C}, \mu_{B|C}$ should also be learnable. The reverse, however, is not true: the total variation is a much stronger distance than the MMD, i.e. there are many cases where the MMD is quite small and the total variation is very large. For instance, the Gaussian-kernel MMD between two nearby point masses will be small, while for total variation and Hellinger it will be maximal.

### C.1 KCI-Power method

Wang et al. (2025), in concurrent parallel work, propose selecting kernel parameters in KCI by maximizing a signal-to-noise ratio, using a scheme closely related to ours. Besides, they jointly tune the kernels $k_A, k_B$, and $k_C$, whereas we optimize only $k_C$; tuning $k_A$ and $k_B$ would require retraining the regressions and thus incur substantially higher computational cost. They recommend a grid search rather than continuous optimization, arguing that estimating conditional mean embeddings introduces intrinsic bias, making gradient-based optimization unstable for improving test power. This limitation is consistent with our observations in Section 5. However, while they attribute the difficulty to continuous optimization, our results show that the real issue is regression accuracy: once the conditional means are well estimated, gradient-based tuning is stable and effective. Although they also note that conditional mean bias can make kernel selection unreliable, our theory explicitly identifies these pitfalls and clarifies the underlying mechanism.

Methodologically, they use the generic unbiased KCI estimator for power maximization, as in (6) (which we use for analytical convenience). In practice, we recommend the HSIC-like unbiased estimator in (20); empirically, this centralized version mitigates conditional mean bias, preventing the selection of kernels that emphasize correlations between regression errors. They also calibrate the null using the Gamma approximation, while we rely on the wild bootstrap. Since Gamma calibration is known to be conservative (Pogodin et al., 2024, Appendix B.2), this accounts for their stronger Type-I control but substantially higher Type-II error. We describe our testing procedure in detail in Appendix H.1

It is also worth noting that Wang et al. present a decomposition of $\mathfrak{C}_{\mathrm{KCI}}$ identical to ours, although they derive it from the earlier formulation (as in Footnote 2) rather than from our first-principles derivation.

## D  Generalization Bound for SNR

For a formal version of Theorem 5.2, we generalize the proof of Liu et al. (2020, Theorem 6) to other second-order $U$-statistics.

Given a set of samples $X_1, \ldots, X_n$ and a function $h$, define

$$U_n := \frac{1}{n(n-1)} \sum_{1 \leq i \neq j \leq n} h(X_i, X_j), \qquad\qquad U := \mathbb{E}\, h(X, X'),$$

$$\sigma^2_{\mathfrak{H}_1,n} := \frac{4}{n^3} \sum_{i=1}^{n} \left( \sum_{j=1}^{n} h(X_i, X_j) \right)^2 - \frac{4}{n^4} \left( \sum_{i,j=1}^{n} h(X_i, X_j) \right)^2, \quad \sigma^2_{\mathfrak{H}_1} := 4\, \mathbb{E}_{X}\Big[ \mathrm{Var}_{X'}[h(X, X') \mid X] \Big],$$

$$\mathrm{SNR}_{n,\lambda} := U_n / \sqrt{\sigma^2_{\mathfrak{H}_1,n} + \lambda}, \qquad\qquad \mathrm{SNR}_{\lambda} := U / \sqrt{\sigma^2_{\mathfrak{H}_1} + \lambda}.$$

and let $\mathrm{SNR} := \mathrm{SNR}_0$. We have constant $\lambda \geq 0$.

Here $U_n$ is the usual second-order $U$-statistic; we assume, without loss of generality, that $h(x, x') = h(x', x)$ for all $x, x'$. We know from Section 5.2.1 of Serfling (1980) (also see Theorem 6.1) that $\mathrm{Var}(U_n) = 4\nu_1/n + \mathcal{O}(1/n^2)$. The estimator $\nu_{1,n}$ follows the biased estimator used by Liu et al.

(2020); while Danica J. Sutherland et al. (2017) used an unbiased variance estimator, the biased estimator is much simpler and also performs better in this setting (Deka and Danica J Sutherland, 2023).

Note that with $X = (A, B, C)$, substituting $h$ in the above formulas with $\hat{h}((A, B, C), (A', B', C'))$ given by

$$k_C(C, C')\langle \phi_A(A) - \hat{\mu}_{A|C}(C), \phi_A(A') - \hat{\mu}_{A|C}(C')\rangle_{\mathcal{H}_A} \langle \phi_B(B) - \hat{\mu}_{B|C}(C), \phi_B(B') - \hat{\mu}_{B|C}(C')\rangle_{\mathcal{H}_B},$$

we have $U = \widehat{\mathrm{KCI}}$, $\sigma^2_{\mathfrak{H}_1}$ is $\hat{\sigma}^2_{\mathfrak{H}_1}$, and SNR is $\widehat{\mathrm{SNR}}$.

**Theorem D.1.** *Let $h_\omega : \mathcal{X} \times \mathcal{X} \to \mathbb{R}$ be a set of functions for each $\omega \in \Omega$ such that:*

- *(i) The $h_\omega$ are uniformly bounded: $\sup_{\omega \in \Omega} \sup_{x, x' \in \mathcal{X}} |h_\omega(x, x')| \leq \rho$ for some $1 \leq \rho < \infty$.*

- *(ii) $\Omega$ is a subset of some $D$-dimensional Banach space, and $\sup_{\omega \in \Omega} \|\omega\| \leq R$.*

- *(iii) The functions are Lipschitz in their parameterization: there is some $L < \infty$ such that for all $x, x' \in \mathcal{X}$ and $\omega, \omega' \in \Omega$, $|h_\omega(x, x') - h_{\omega'}(x, x')| \leq L\|\omega - \omega'\|$.*

*Use $U_n^{(\omega)}$ and similar to denote the quantities defined above with the function $h_\omega$. Let $\bar{\Omega}_s \subseteq \Omega$ be a set of parameters for which $\sigma^{(\omega)}_{\mathfrak{H}_1} \geq s$. Take $\lambda = \ell n^{-1/3}$. Then, with probability at least $1 - \delta$,*

$$\sup_{\omega \in \bar{\Omega}_s} |\mathrm{SNR}^{(\omega)}_{n,\lambda} - \mathrm{SNR}^{(\omega)}| \leq \frac{\rho}{s^2 n^{1/3}} \left[ \frac{\ell}{2s} + \left[ \frac{448\rho}{\sqrt{\ell}} + \frac{2s}{n^{1/6}} \right] \left[ L + \sqrt{2 \log \frac{4}{\delta} + 2D \log(4R\sqrt{n})} \right] + \frac{72\rho^2}{\sqrt{\ell n}} \right].$$

*Thus, treating $\rho$ and $\ell$ as constants, we have that*

$$\sup_{\omega \in \bar{\Omega}_s} |\mathrm{SNR}^{(\omega)}_{n,\lambda} - \mathrm{SNR}^{(\omega)}| = \mathcal{O}\left( \frac{1}{s^2 n^{1/3}} \left[ \frac{1}{s} + \left(1 + \frac{s}{n^{1/6}}\right) \left[ L + \sqrt{D \log(Rn) + \log \frac{1}{\delta}} \right] \right] \right).$$

*This further implies that if $\mathrm{SNR}^{(\omega)}$ has a unique maximizer $\omega^* \in \bar{\Omega}_s$, the sequence of empirical minimizers of $\mathrm{SNR}^{(\omega)}_{n,\lambda=\ell n^{-\frac{1}{3}}}$ converges in probability to $\omega^*$.*

The assumptions in Theorem D.1 agree with those of Liu et al. (2020). Their Appendix A.4's bounds on $L$ directly apply to the $\hat{h}$ of $\widehat{\mathrm{KCI}}$ if we only consider changing $k_C$, as we do in our experiments. These techniques could be readily adapted to changing other parameters, whether $k_A$ or $k_B$ (if the regressions are also updated appropriately) or parameters inside $\hat{\mu}_{A|C}$ and $\hat{\mu}_{B|C}$. We emphasize, however, that doing so only increases $\widehat{\mathrm{SNR}}$; any of these operations could increase the probability of rejecting the null under the alternative, but they will *also* increase the probability of rejecting the null under the null, further inflating Type-I error.

*Proof.* Let $\sigma^2_{\mathfrak{H}_1,n,\lambda} = \sigma^2_{\mathfrak{H}_1,n} + \lambda$ and $\sigma^2_{\mathfrak{H}_1,\lambda} = \sigma^2_{\mathfrak{H}_1} + \lambda$. We begin with the decomposition

$$\sup_{\omega \in \bar{\Omega}_s} |\mathrm{SNR}^{(\omega)}_{n,\lambda} - \mathrm{SNR}^{(\omega)}| = \sup_{\omega \in \bar{\Omega}_s} \left| \frac{U_n^{(\omega)}}{\sigma^{(\omega)}_{\mathfrak{H}_1,n,\lambda}} - \frac{U^{(\omega)}}{\sigma^{(\omega)}_{\mathfrak{H}_1}} \right|$$

$$\leq \sup_{\omega \in \bar{\Omega}_s} \left| \frac{U_n^{(\omega)}}{\sigma^{(\omega)}_{\mathfrak{H}_1,n,\lambda}} - \frac{U_n^{(\omega)}}{\sigma^{(\omega)}_{\mathfrak{H}_1,\lambda}} \right| + \sup_{\omega \in \bar{\Omega}_s} \left| \frac{U_n^{(\omega)}}{\sigma^{(\omega)}_{\mathfrak{H}_1,\lambda}} - \frac{U_n^{(\omega)}}{\sigma^{(\omega)}_{\mathfrak{H}_1}} \right| + \sup_{\omega \in \bar{\Omega}_s} \left| \frac{U_n^{(\omega)}}{\sigma^{(\omega)}_{\mathfrak{H}_1}} - \frac{U^{(\omega)}}{\sigma^{(\omega)}_{\mathfrak{H}_1}} \right|.$$

Now notice that $|U_n^\omega| \leq \rho$, $\sigma^{(\omega)}_{\mathfrak{H}_1,\lambda} \geq \sqrt{s^2 + \lambda} \geq s$, and $\sigma^{(\omega)}_{\mathfrak{H}_1,n,\lambda} \geq \sqrt{\lambda}$. Hence the first term is

$$\sup_{\omega \in \bar{\Omega}_s} \left| \frac{U_n^{(\omega)}}{\sigma^{(\omega)}_{\mathfrak{H}_1,n,\lambda}} - \frac{U_n^{(\omega)}}{\sigma^{(\omega)}_{\mathfrak{H}_1,\lambda}} \right| = \sup_{\omega \in \bar{\Omega}_s} |U_n^{(\omega)}| \frac{1}{\sigma^{(\omega)}_{\mathfrak{H}_1,n,\lambda}} \frac{1}{\sigma^{(\omega)}_{\mathfrak{H}_1,\lambda}} \frac{|(\sigma^{(\omega)}_{\mathfrak{H}_1,n,\lambda})^2 - (\sigma^{(\omega)}_{\mathfrak{H}_1,\lambda})^2|}{\sigma^{(\omega)}_{\mathfrak{H}_1,n,\lambda} + \sigma^{(\omega)}_{\mathfrak{H}_1,\lambda}}$$

$$\leq \frac{\rho}{\sqrt{\lambda}\sqrt{s^2 + \lambda}(\sqrt{s^2 + \lambda} + \sqrt{\lambda})} \sup_{\omega \in \bar{\Omega}_s} |(\sigma^{(\omega)}_{\mathfrak{H}_1,n,\lambda})^2 - (\sigma^{(\omega)}_{\mathfrak{H}_1,\lambda})^2|$$

$$\leq \frac{\rho}{s^2\sqrt{\lambda}} \sup_{\omega \in \bar{\Omega}_s} |(\sigma^{(\omega)}_{\mathfrak{H}_1,n,\lambda})^2 - (\sigma^{(\omega)}_{\mathfrak{H}_1,\lambda})^2|,$$

the second is

$$\sup_{\omega\in\bar\Omega_s}\left|\frac{U_n^{(\omega)}}{\sigma_{\mathfrak{H}_1,\lambda}^{(\omega)}}-\frac{U_n^{(\omega)}}{\sigma_{\mathfrak{H}_1}^{(\omega)}}\right|=\sup_{\omega\in\bar\Omega_s}|U_n^{(\omega)}|\,\frac{1}{\sigma_{\mathfrak{H}_1,\lambda}^{(\omega)}}\frac{1}{\sigma_{\mathfrak{H}_1}^{(\omega)}}\left|\frac{(\sigma_{\mathfrak{H}_1,\lambda}^{(\omega)})^2-(\sigma_{\mathfrak{H}_1}^{(\omega)})^2}{\sigma_{\mathfrak{H}_1,\lambda}^{(\omega)}+\sigma_{\mathfrak{H}_1}^{(\omega)}}\right|\le\frac{\rho\lambda}{2s^3},$$

and the third is

$$\sup_{\omega\in\bar\Omega_s}\left|\frac{U_n^{(\omega)}}{\sigma_{\mathfrak{H}_1}^{(\omega)}}-\frac{U^{(\omega)}}{\sigma_{\mathfrak{H}_1}^{(\omega)}}\right|=\sup_{\omega\in\bar\Omega_s}\frac{1}{\sigma_{\mathfrak{H}_1}^{(\omega)}}\left|U_n^{(\omega)}-U^{(\omega)}\right|\le\frac{1}{s}\sup_{\omega\in\bar\Omega_s}\left|U_n^{(\omega)}-U^{(\omega)}\right|.$$

Thus we have reduced to needing uniform convergence of $U_n$ and $\sigma_{\mathfrak{H}_1,n,\lambda}^2$.

Propositions 15 and 16 of Liu et al. (2020) show this, up to replacing their $\nu$ with our $\rho/4$, their $R_\Omega$ with our $R$, and their $L_k$ with our $L/4$; this can be seen by inspecting the proofs. The results become

$$\Pr\left(\sup_{\omega\in\Omega}\left|U_n^{(\omega)}-U^{(\omega)}\right|\le\frac{2}{\sqrt{n}}\left[\rho\sqrt{2\log\frac{2}{\delta}+2D\log(4R\sqrt{n})}+L\right]\right)\ge1-\delta$$

$$\Pr\left(\sup_{\omega\in\Omega}\left|(\sigma_{\mathfrak{H}_1,n,\lambda}^{(\omega)})^2-(\sigma_{\mathfrak{H}_1,\lambda}^{(\omega)})^2\right|\le\frac{64}{\sqrt{n}}\left[7\sqrt{2\log\frac{2}{\delta}+2D\log(4R\sqrt{n})}+\frac{9\rho^2}{8\sqrt{n}}+\frac{1}{2}L\rho\right]\right)\ge1-\delta.$$

Combining the results, it holds with probability at least $1-\delta$ that the worst-case error $\sup_{\omega\in\bar\Omega_s}|\mathrm{SNR}_{n,\lambda}^{(\omega)}-\mathrm{SNR}^{(\omega)}|$ is at most

$$\frac{\rho\lambda}{2s^3}+\left[\frac{2\rho}{s\sqrt{n}}+\frac{448\rho}{s^2\sqrt{\lambda n}}\right]\sqrt{2\log\frac{4}{\delta}+2D\log(4R\sqrt{n})}+\left[\frac{2}{s\sqrt{n}}+\frac{32\rho^2}{s^2\sqrt{\lambda n}}\right]L+\frac{72\rho^3}{s^2n\sqrt{\lambda}}.$$

Plugging in $\lambda=\ell n^{-\frac13}$ yields

$$\frac{\rho\ell}{2s^3n^{\frac13}}+\left[\frac{2\rho}{s\sqrt{n}}+\frac{448\rho}{\sqrt{\ell}s^2n^{\frac13}}\right]\sqrt{2\log\frac{4}{\delta}+2D\log(4R\sqrt{n})}+\left[\frac{2}{s\sqrt{n}}+\frac{32\rho^2}{\sqrt{\ell}s^2n^{\frac13}}\right]L+\frac{72\rho^3}{\sqrt{\ell}s^2n^{\frac56}}.$$

We can use our assumption $\rho\ge1$ and that $448>32$ to get a slightly looser but simpler upper bound of

$$\frac{\rho\ell}{2s^3n^{1/3}}+\left[\frac{2\rho}{s\sqrt{n}}+\frac{448\rho^2}{\sqrt{\ell}s^2n^{1/3}}\right]\left[L+\sqrt{2\log\frac{4}{\delta}+2D\log(4R\sqrt{n})}\right]+\frac{72\rho^3}{\sqrt{\ell}s^2n^{5/6}},$$

which reduces to the result in the theorem statement.

The final result is a standard consequence of the prior statement, as in Corollary 12 of Liu et al. (2020). $\qquad\square$

# E    U-Statistic Moments for Hilbert Space Kernels

**Theorem 6.1.** *Let $h(X,X')=\langle\phi_h(X),\phi_h(X')\rangle$ with mean embedding $\mu_h=\mathbb{E}_X\,\phi_h(X)$ and the centered covariance operator $\mathfrak{C}_h=\mathbb{E}_X[\phi_h(X)\otimes\phi_h(X)]-\mu_h\otimes\mu_h$. Define $\nu_1=\langle\mu_h,\mathfrak{C}_h\mu_h\rangle$ and $\nu_2=\|\mathfrak{C}_h\|_{\mathrm{HS}}^2$. The corresponding $U$-statistic satisfies*

$$U_n=\frac{1}{n(n-1)}\sum_{1\le i\neq j\le n}h(X_i,X_j),\quad\mathbb{E}[U_n]=U=\|\mu_h\|^2,\quad\mathrm{Var}(U_n)=\frac{4}{n}\nu_1+\frac{2}{n(n-1)}\nu_2.$$

*Proof.* $U_n$ is the definition of a second-order $U$-statistic. We have that

$$\begin{aligned}\mathbb{E}\,U_n&=\mathop{\mathbb{E}}_{X,X'}h(X,X')\\&=\mathop{\mathbb{E}}_{X,X'}\langle\phi_h(X),\phi_h(X')\rangle\\&=\langle\mathop{\mathbb{E}}_X\phi_h(X),\mathop{\mathbb{E}}_{X'}\phi_h(X)\rangle=\langle\mu_h,\mu_h\rangle=\|\mu_h\|^2\end{aligned}$$

when $\mu_h$ exists in the Bochner sense, $\mathbb{E}\|\phi_h(X)\|<\infty$.

For the variance, it is a standard result that (e.g. Serfling, 1980, Section 5.2.1):

$$\mathrm{Var}(U_n) = \frac{4(n-2)}{n(n-1)} \, \underset{X}{\mathrm{Var}} \Big[ \underset{X'|X}{\mathbb{E}} [\, h(X, X') \mid X \,] \Big] + \frac{2}{n(n-1)} \, \mathrm{Var} \Big[ h(X, X') \Big].$$

Using the law of total variance,

$$\mathrm{Var}\Big[ h(X, X') \Big] = \underset{X}{\mathrm{Var}} \Big[ \underset{X'|X}{\mathbb{E}} [\, h(X, X') \mid X \,] \Big] + \underset{X}{\mathbb{E}} \Big[ \underset{X'|X}{\mathrm{Var}} [\, h(X, X') \mid X \,] \Big]$$

and so

$$\mathrm{Var}(U_n) = \frac{4n-6}{n(n-1)} \, \underset{X}{\mathrm{Var}} \Big[ \underset{X'|X}{\mathbb{E}} [\, h(X, X') \mid X \,] \Big] + \frac{2}{n(n-1)} \, \underset{X}{\mathbb{E}} \Big[ \underset{X'|X}{\mathrm{Var}} [\, h(X, X') \mid X \,] \Big].$$

We can now compute that

$$\underset{X'|X}{\mathbb{E}} [h(X, X') \mid X] = \langle \phi_h(X), \mu_h \rangle$$

$$\begin{aligned}
\underset{X}{\mathrm{Var}} \Big[ \underset{X'|X}{\mathbb{E}} [h(X, X') \mid X] \Big] &= \underset{X}{\mathbb{E}} \Big[ \big( \underset{X'|X}{\mathbb{E}} [h(X, X') \mid X] \big)^2 \Big] - \Big( \underset{X}{\mathbb{E}} \underset{X'|X}{\mathbb{E}} [h(X, X') \mid X] \Big)^2 \\
&= \underset{X}{\mathbb{E}} \langle \phi_h(X), \mu_h \rangle^2 - \langle \mu_h, \mu_h \rangle^2 \\
&= \underset{X}{\mathbb{E}} \langle \mu_h, \phi_h(X) \rangle \langle \phi_h(X), \mu_h \rangle - \langle \mu_h, \mu_h \rangle \langle \mu_h, \mu_h \rangle \\
&= \underset{X}{\mathbb{E}} \langle \mu_h, \big( \phi_h(X) \otimes \phi_h(X) - \mu_h \otimes \mu_h \big) \mu_h \rangle \\
&= \Big\langle \mu_h, \underset{X}{\mathbb{E}} \big[ \phi_h(X) \otimes \phi_h(X) - \mu_h \otimes \mu_h \big] \mu_h \Big\rangle \\
&= \langle \mu_h, \mathfrak{C}_h \mu_h \rangle = \nu_1.
\end{aligned}$$

The remaining term is given by

$$\begin{aligned}
\underset{X'|X}{\mathrm{Var}} \Big[ h(X, X') \mid X \Big] &= \underset{X'|X}{\mathbb{E}} \Big[ h(X, X')^2 \mid X \Big] - \Big( \underset{X'|X}{\mathbb{E}} \Big[ h(X, X') \mid X \Big] \Big)^2 \\
&= \underset{X'|X}{\mathbb{E}} \langle \phi_h(X), \phi_h(X') \rangle \langle \phi_h(X'), \phi_h(X) \rangle - \Big( \underset{X'|X}{\mathbb{E}} \langle \phi_h(X), \phi_h(X') \rangle \Big)^2 \\
&= \underset{X'|X}{\mathbb{E}} \langle \phi_h(X), \phi_h(X') \rangle \langle \phi_h(X'), \phi_h(X) \rangle - \langle \phi_h(X), \mu_h \rangle \langle \phi_h(X), \mu_h \rangle \\
&= \Big\langle \phi_h(X), \big( \underset{X'|X}{\mathbb{E}} \phi_h(X') \otimes \phi_h(X') - \mu_h \otimes \mu_h \big) \phi_h(X) \Big\rangle \\
&= \langle \phi_h(X), \mathfrak{C}_h \phi_h(X) \rangle \\
\underset{X}{\mathbb{E}} \underset{X'|X}{\mathrm{Var}} \Big[ h(X, X') \mid X \Big] &= \underset{X}{\mathbb{E}} \langle \phi_h(X), \mathfrak{C}_h \phi_h(X) \rangle \\
&= \underset{X}{\mathbb{E}} \langle \phi_h(X) \otimes \phi_h(X), \mathfrak{C}_h \rangle_{\mathrm{HS}} \\
&= \Big\langle \underset{X}{\mathbb{E}} \phi_h(X) \otimes \phi_h(X), \mathfrak{C}_h \Big\rangle_{\mathrm{HS}} \\
&= \langle \mathfrak{C}_h + \mu_h \otimes \mu_h, \mathfrak{C}_h \rangle_{\mathrm{HS}} = \nu_1 + \nu_2.
\end{aligned}$$

Combining, we find that

$$\mathrm{Var}(U_n) = \frac{4n-6}{n(n-1)} \nu_1 + \frac{2}{n(n-1)} (\nu_1 + \nu_2) = \frac{4}{n} \nu_1 + \frac{2}{n(n-1)} \nu_2. \qquad \square$$

# F   Analytical Example

## F.1   With correct regressions

**KCI as an expectation under a bivariate Gaussian.**   Under the assumption of linear kernels $\phi_A(a) = a$ and $\phi_B(b) = b$, the conditional cross-covariance operator with correct regressions can be

written as:

$$\mathfrak{C}_{AB|C} = \underset{AB|C}{\mathbb{E}} \big[ (A - \mu_{A|C}(C))(B - \mu_{B|C}(C)) \mid C \big]$$
$$= \underset{AB|C}{\mathbb{E}} \big[ (A - f_A(C))(B - f_B(C)) \mid C \big]$$
$$= \tau^2 E_{AB|C} \big[ r_A \, r_B \mid C \big]$$
$$= \tau^2 \gamma(C)$$

Since $C, C' \sim \mathcal{N}(0,1)$ independently, and $k_C(c,c') = \exp\left(-\frac{(c-c')^2}{2\ell_C^2}\right)$, then the KCI statistic becomes:

$$\text{KCI} = \tau^4 \underset{C,C'}{\mathbb{E}} \big[ k_C(C,C')\gamma(C)\gamma(C') \big].$$

Using the fact that $C, C'$ are independent standard Gaussians, this becomes:

$$\text{KCI} = \tau^4 \iint \frac{1}{2\pi} \exp\left(-\frac{c^2 + c'^2}{2}\right) \exp\left(-\frac{(c-c')^2}{2\ell_C^2}\right) \gamma(c)\gamma(c')dcdc'$$
$$= \tau^4 \iint \frac{1}{2\pi} \exp\left(-\frac{(\ell_C^2+1)c^2 - 2cc' + (\ell_C^2+1)c'^2}{2\ell_C^2}\right) \gamma(c)\gamma(c')dcdc',$$

Define the vector $\mathbf{x} = \begin{bmatrix} c \\ c' \end{bmatrix}$, and write the integrand as a bivariate Gaussian density with covariance matrix $\mathbf{\Sigma}$. That is,

$$\text{KCI} = \tau^4 \sqrt{\frac{\ell_C^2}{\ell_C^2 + 2}} \int_{\mathbb{R}^2} \phi_{\mathbf{\Sigma}}(c,c') \, \gamma(c) \, \gamma(c') \, dcdc',$$

where $\phi_{\mathbf{\Sigma}}$ denotes the bivariate normal density with zero mean and covariance matrix

$$\mathbf{\Sigma} = \begin{bmatrix} \frac{\ell_C^2+1}{\ell_C^2+2} & \frac{1}{\ell_C^2+2} \\ \frac{1}{\ell_C^2+2} & \frac{\ell_C^2+1}{\ell_C^2+2} \end{bmatrix}, \quad \det(\mathbf{\Sigma}) = \frac{\ell_C^2}{\ell_C^2 + 2}.$$

We may thus express:

$$\text{KCI} = \tau^4 \sqrt{\frac{\ell_C^2}{\ell_C^2 + 2}} \underset{(X,X') \sim \mathcal{N}_{\ell_C}}{\mathbb{E}} \big[ \gamma(X)\gamma(X') \big],$$

with auxiliary variables $(X, X') \sim \mathcal{N}_{\ell_C} := \mathcal{N}\left( \begin{bmatrix} 0 \\ 0 \end{bmatrix}, \begin{bmatrix} 1 - \frac{1}{\ell_C^2+2} & \frac{1}{\ell_C^2+2} \\ \frac{1}{\ell_C^2+2} & 1 - \frac{1}{\ell_C^2+2} \end{bmatrix} \right).$

**Exact expression for** KCI   We can analytically compute both the population KCI value and its variance to generate the theoretical curve shown in Figure 2. Under the alternative hypothesis, suppose the conditional dependence takes the form $\gamma(X) = \sin(\beta X)$. Then the KCI statistic becomes:

$$\text{KCI} = \tau^4 \sqrt{\frac{\ell_C^2}{\ell_C^2 + 2}} \underset{(X,X') \sim \mathcal{N}_{\ell_C}}{\mathbb{E}} \big[ \sin(\beta X)\sin(\beta X') \big]$$
$$= \frac{\tau^4}{2} \sqrt{\frac{\ell_C^2}{\ell_C^2 + 2}} \underset{(X,X') \sim \mathcal{N}_{\ell_C}}{\mathbb{E}} \big[ \cos(\beta(X - X')) - \cos(\beta(X + X')) \big],$$

Now note that $X - X'$ and $X + X'$ are linear functions of a jointly Gaussian vector and hence are Gaussian themselves. Since $(X, X') \sim \mathcal{N}(0, \mathbf{\Sigma})$, the random variables $Z_1 = X - X'$, $Z_2 = X + X'$ are zero-mean and have variances:

$$\text{Var}(Z_1) = 2\left(1 - \frac{2}{\ell_C^2 + 2}\right), \qquad \text{Var}(Z_2) = 2.$$

We now compute the expectations using the identity for the cosine of a Gaussian: $\mathbb{E}[\cos(\beta Z)] = \exp\left(-\frac{1}{2}\beta^2 \operatorname{Var}(Z)\right)$. Thus,

$$\mathbb{E}_{X,X'}\left[\cos\left(\beta(X - X')\right)\right] = \exp\left(-\beta^2\left(1 - \frac{2}{\ell_C^2 + 2}\right)\right),$$

$$\mathbb{E}_{X,X'}\left[\cos\left(\beta(X + X')\right)\right] = \exp\left(-\beta^2\right).$$

Substituting into the expression for KCI, we obtain:

$$\mathrm{KCI} = \frac{\tau^4}{2}\sqrt{\frac{\sigma_C}{\sigma_C + 2}}\left(\exp\left(-\beta^2\left(1 - \frac{2}{\sigma_C + 2}\right)\right) - \exp\left(-\beta^2\right)\right)$$

$$= \frac{\tau^4}{2}\exp(-\beta^2)\sqrt{\frac{\sigma_C}{\sigma_C + 2}}\left(\exp\left(\frac{2\beta^2}{\sigma_C + 2}\right) - 1\right).$$

**Exact expression of variance of** $\mathrm{KCI}_n$**.** The variance of the U-statistic $\mathrm{KCI}_n$ can be decomposed into three components, as described in Appendix E. We now provide exact expressions for each term under the alternative hypothesis.

$$v_{\mathrm{c}} := \underset{ABC}{\mathbb{E}}\left[\left(\underset{A'B'C'}{\mathbb{E}}[h(ABC, A'B'C') \mid ABC]\right)^2\right]$$

$$= \underset{ABC}{\mathbb{E}}\left[\left(\underset{A'B'C'}{\mathbb{E}}[\tau^4 k_C(C,C')r_A r_B r_A' r_B' \mid ABC]\right)^2\right]$$

$$= \tau^8\,\underset{C}{\mathbb{E}}\left[\underset{AB}{\mathbb{E}}[r_A^2 r_B^2 \mid C]\left(\underset{C'}{\mathbb{E}}\left[k_C(C,C')\underset{A'B'\mid C'}{\mathbb{E}}[r_A' r_B' \mid C']\right]\right)^2\right]$$

$$= \tau^8\,\underset{C}{\mathbb{E}}\left[(1 + 2\gamma^2(\beta C))\left(\underset{C'}{\mathbb{E}}[k_C(C,C')\gamma(\beta C')]\right)^2\right]$$

$$= \frac{\tau^8 \ell_C^2}{\sqrt{(\ell_C^2 + 1)(\ell_C^2 + 3)}}\,\underset{X\sim\mathcal{N}(0,\frac{\ell_C^2+1}{\ell_C^2+3})}{\mathbb{E}}\left[(1 + 2\gamma^2(\beta X))\left(\underset{X'\sim\mathcal{N}(\frac{X}{\ell_C^2+1},\frac{\ell_C^2}{\ell_C^2+1})}{\mathbb{E}}[\gamma(\beta X')]\right)^2\right]$$

$$= \frac{\tau^8 \ell_C^2 \exp(-\frac{\beta^2 \ell_C^2}{\ell_C^2+1})}{\sqrt{(\ell_C^2 + 1)(\ell_C^2 + 3)}}\left(1 - \exp\left(-\frac{2\beta^2}{(\ell_C^2+1)(\ell_C^2+3)}\right) - \frac{1}{2}\exp\left(-\frac{2\beta^2(\ell_C^2+1)}{\ell_C^2+3}\right)\right.$$

$$\left. + \frac{1}{4}\exp\left(-\frac{2\beta^2(\ell_C^2+2)^2}{(\ell_C^2+1)(\ell_C^2+3)}\right) + \frac{1}{4}\exp\left(-\frac{2\beta^2 \ell_C^4}{(\ell_C^2+1)(\ell_C^2+3)}\right)\right).$$

Besides,

$$v_{\mathrm{m}} := \left(\underset{ABC,A'B'C'}{\mathbb{E}}[h(ABC, A'B'C')]\right)^2 = \mathrm{KCI}^2.$$

Also,

$$v_{\mathrm{s}} := \mathbb{E}\left[h^2(ABC, A'B'C')\right]$$

$$= \tau^8\,\mathbb{E}\left[k_C^2(C,C')r_A^2 r_B^2 r_A'^2 r_B'^2\right]$$

$$= \tau^8\,\underset{C,C'}{\mathbb{E}}\left[k_C^2(C,C')\underset{A,B}{\mathbb{E}}[r_A^2 r_B^2 \mid C]\underset{A',B'}{\mathbb{E}}[r_A'^2 r_B'^2 \mid C']\right]$$

$$= \tau^8\,\underset{C,C'}{\mathbb{E}}\left[\exp\left(\frac{(C-C')^2}{\ell_C^2}\right)(1 + 2\gamma^2(C))(1 + 2\gamma^2(C'))\right]$$

$$= \tau^8\sqrt{\frac{\ell_C^2}{\ell_C^2 + 4}}\left(4 - 2\exp\left(-\frac{2\beta^2(\ell_C^2+2)}{\ell_C^2+4}\right) + \exp\left(-\frac{2\beta^2 \ell_C^2}{\ell_C^2+2}\right)\right.$$

$$\left.\cdot\left(-2\exp\left(\frac{-8\beta^2}{(\ell_C^2+2)(\ell_C^2+4)}\right) + \frac{1}{2}\exp\left(\frac{-2\beta^2(\ell_C^2+4)}{\ell_C^2+2}\right) + \frac{1}{2}\exp\left(\frac{-2\beta^2 \ell_C^4}{(\ell_C^2+2)(\ell_C^2+4)}\right)\right)\right).$$

Therefore, the variance can be obtained by combining those three terms together:

$$\operatorname{Var}(\mathrm{KCI}_n) = \frac{(4n-8)v_{\mathrm{c}} - (4n-6)v_{\mathrm{m}} + 2v_{\mathrm{s}}}{n(n-1)}.$$

## F.2 With regression errors

Suppose the conditional mean embeddings have errors, that $\widehat{\mu}_{A|C} = \mu_{A|C} + \Delta_{A|C}$ and $\widehat{\mu}_{B|C} = \mu_{B|C} + \Delta_{B|C}$, where $\Delta_{A|C}$ and $\Delta_{B|C}$ denote the respective regression errors. Then the conditional cross-covariance operator becomes:

$$
\begin{aligned}
\widehat{\mathfrak{C}}_{AB|C}(C) &= \mathop{\mathbb{E}}_{AB|C}\left[(A - \widehat{\mu}_{A|C}(C))(B - \widehat{\mu}_{B|C}(C)) \mid C\right] \\
&= \mathop{\mathbb{E}}_{AB|C}\left[(A - f_A(C) - \Delta_{A|C}(C))(B - f_B(C) - \Delta_{B|C}(C)) \mid C\right] \\
&= E_{AB|C}\left[(\tau r_A - \Delta_{A|C}(C))(\tau r_B - \Delta_{B|C}(C)) \mid C\right] \\
&= \tau^2 \gamma(C) + \Delta_{A|C}(C)\Delta_{B|C}(C).
\end{aligned}
$$

The final equality follows from the assumption that the regression estimates are obtained using an independent training set, and are thus independent of the test-time noise in $r_A$ and $r_B$. Consequently, the cross terms involving $r_A \Delta_{B|C}(C)$ and $r_B \Delta_{A|C}(C)$ have zero conditional expectation.

Thus, under the null hypothesis ($A \perp\!\!\!\perp B \mid C$), the KCI with noisy conditional means becomes

$$
\widehat{\mathrm{KCI}} = \mathop{\mathbb{E}}_{CC'}\left[k_C(C, C')\Delta_{A|C}(C)\Delta_{B|C}(C)\Delta_{A|C}(C')\Delta_{B|C}(C')\right].
$$

Under the null hypothesis, the variance of the U-statistic $\widehat{\mathrm{KCI}}_n$ decomposes into the following three components, noting that $\mathbb{E}_{A|C}[r_A{}^2] = 1$ and $\mathbb{E}_{B|C}[r_B{}^2] = 1$:

$$
\begin{aligned}
v_{\mathrm{c}} &= \mathop{\mathbb{E}}_{ABC}\left[\left(\mathop{\mathbb{E}}_{A'B'C'}\left[k_C(C, C')(\tau r_A - \Delta_{A|C}(C))(\tau r_B - \Delta_{B|C}(C))\right.\right.\right. \\
&\qquad\qquad\qquad\qquad \left.\left.\left.\cdot (\tau r_A{}' - \Delta_{A|C}(C'))(\tau r_B{}' - \Delta_{B|C}(C'))\right]\right)^2\right] \\
&= \mathop{\mathbb{E}}_{ABC}\left[(\tau r_A - \Delta_{A|C}(C))^2(\tau r_B - \Delta_{B|C}(C))^2\left(\mathop{\mathbb{E}}_{C'}\left[k_C(C, C')\Delta_{A|C}(C')\Delta_{B|C}(C')\right]\right)^2\right] \\
&= \mathop{\mathbb{E}}_{C}\left[(\tau^2 + \Delta_{A|C}{}^2(C))(\tau^2 + \Delta_{B|C}{}^2(C))\left(\mathop{\mathbb{E}}_{C'}\left[k_C(C, C')\Delta_{A|C}(C')\Delta_{B|C}(C')\right]\right)^2\right].
\end{aligned}
$$

$$
v_{\mathrm{m}} = \left(\mathop{\mathbb{E}}_{CC'}\left[k_C(C, C')\Delta_{A|C}(C)\Delta_{B|C}(C)\Delta_{A|C}(C')\Delta_{B|C}(C')\right]\right)^2.
$$

$$
\begin{aligned}
v_{\mathrm{s}} &= \mathbb{E}\left[k_C{}^2(C, C')(\tau r_A - \Delta_{A|C}(C))^2(\tau r_B - \Delta_{B|C}(C))^2(\tau r_A{}' - \Delta_{A|C}(C'))^2(\tau r_B{}' - \Delta_{B|C}(C'))^2\right] \\
&= \mathbb{E}\left[k_C{}^2(C, C')(\tau^2 + \Delta_{A|C}{}^2(C))(\tau^2 + \Delta_{B|C}{}^2(C))(\tau^2 + \Delta_{A|C}{}^2(C'))(\tau^2 + \Delta_{B|C}{}^2(C'))\right].
\end{aligned}
$$

Combining these three terms we get:

$$
\mathrm{Var}(\widehat{\mathrm{KCI}}_n) = \frac{(4n - 8)v_{\mathrm{c}} - (4n - 6)v_{\mathrm{m}} + 2v_{\mathrm{s}}}{n(n - 1)}.
$$

## F.3 Complex conditional dependence scenario (3-dimensional $C$)

We now extend our motivating example to a more complex setting by considering a three-dimensional conditioning variable $C = (C_1, C_2, C_3)$. Specifically, we define random variables:

$$
C \sim \mathcal{N}(0, I_3), \quad A = f_A(e_A^\top C) + \tau r_A, \quad B = f_B(e_B^\top C) + \tau r_B,
$$

where $e_A$ and $e_B$ are indicator vectors selecting specific dimensions of (one entry equals 1, others 0), determining which dimension influences each variable. The additive noise terms $(r_A, r_B)$ are conditionally dependent via on $\gamma(e_C^\top C)$, as previously defined.

We employ a generalized Gaussian kernel for $C$ with dimension-specific (squared) lengthscales $\ell_{C_i}^2$, as commonly implemented in libraries such as `sklearn`: $k_C(C, C') = \exp\left(-\sum_{i=1}^{3} \frac{(C_i - C_i')^2}{2\ell_{C_i}^2}\right)$.

We assume regression models are trained specifically on the relevant dimensions (as selected by $e_A$ and $e_B$), effectively ignoring irrelevant or noisy dimensions. Thus, $\Delta_{A|C}(e_A{}^\top C) = \widehat{\mu}_{A|C}(e_A{}^\top C) - \mu_{A|C}(e_A{}^\top C)$ and $\Delta_{B|C}(e_B{}^\top C) = \widehat{\mu}_{B|C}(e_B{}^\top C) - \mu_{B|C}(e_B{}^\top C)$ depend only on the dimensions directly influencing $A$ and $B$. The noisy KCI statistic $\widehat{\text{KCI}}$, using linear kernels for $A$ and $B$, becomes:

$$\widehat{\text{KCI}}$$
$$= \mathbb{E}\left[k_C(C, C')\big(\tau^2\gamma(e_C{}^\top C) + \xi(e_A{}^\top C, e_B{}^\top C)\big)\big(\tau^2\gamma(e_C{}^\top C') + \xi(e_A{}^\top C', e_B{}^\top C')\big)\right]$$
$$= \left(\prod_{i=1}^{3}\sqrt{\frac{\ell_{Ci}^2}{\ell_{Ci}^2 + 2}}\right)\mathbb{E}\Big[\tau^4\,\gamma(X_C)\gamma(X_C') + \xi(X_A, X_B)\,\gamma(X_C')$$
$$+ \xi(X_A', X_B')\,\gamma(X_C) + \xi(X_A, X_B)\,\xi(X_A', X_B')\Big]$$

where $X_C = e_C{}^\top X$, $\xi(X_A, X_B) = \Delta_{A|C}(e_A{}^\top X)\Delta_{B|C}(e_B{}^\top X)$. Similar to Appendix F.1, $X$ is an auxiliary variable, and for $i = 1, 2, 3$, we have $X_i, X_i' \sim \mathcal{N}_{\ell_{Ci}}$.

Kernel lengthscale selection and regression errors critically influence test performance. We discuss two illustrative scenarios:

**Scenario 1: Shared-coordinate dependence ($e_A = e_B = e_C$):** . When $A$, $B$, and their conditional dependence all rely on the same coordinate (e.g., $C_1$). The KCI is

$$\widehat{\text{KCI}} = \left(\prod_{i=1}^{3}\sqrt{\frac{\ell_{Ci}^2}{\ell_{Ci}^2 + 2}}\right)\mathbb{E}\left[\tau^4\gamma(X_1)\gamma(X_1') + \xi(X_1, X_1)\gamma(X_1') + \xi(X_1', X_1')\gamma(X_1) + \xi(X_1, X_1)\xi(X_1', X_1')\right].$$

Specifically, under the null hypothesis, the regression error will leak "dependence" into the test,

$$\widehat{\text{KCI}} = \left(\prod_{i=1}^{3}\sqrt{\frac{\ell_{Ci}^2}{\ell_{Ci}^2 + 2}}\right)\mathbb{E}\left[\xi(X_1, X_1)\xi(X_1', X_1')\right].$$

Explicitly, this is:

$$\widehat{\text{KCI}} = \left(\prod_{i=1}^{3}\sqrt{\frac{\ell_{Ci}^2}{\ell_{Ci}^2 + 2}}\right)\mathbb{E}\left[\Delta_{A|C}(X_1)\Delta_{A|C}(X_1')\,\Delta_{B|C}(X_1)\Delta_{B|C}(X_1')\right].$$

**Scenario 2: Separate-coordinate dependence (distinct $e_A, e_B, e_C$)** . When $A, B$ and their conditional dependence each utilize distinct coordinates (e.g., $C_1, C_2, C_3$ respectively), because of the independence between $(X_A, X_B)$ and $X_C$, the KCI becomes:

$$\widehat{\text{KCI}} = \left(\prod_{i=1}^{3}\sqrt{\frac{\ell_{Ci}^2}{\ell_{Ci}^2 + 2}}\right)\mathbb{E}\left[\tau^4\gamma(X_3)\gamma(X_3') + \xi(X_1, X_2)\xi(X_1', X_2').\right].$$

where we can further decompose the noise

$$\mathbb{E}[\xi(X_2, X_3)\xi(X_2', X_3')] = \mathbb{E}\left[\Delta_{A|C}(X_2)\Delta_{A|C}(X_2')\right]\,\mathbb{E}\left[\Delta_{B|C}(X_3)\Delta_{B|C}(X_3')\right]$$

Under the null hypothesis, the KCI becomes

$$\widehat{\text{KCI}} = \left(\prod_{i=1}^{3}\sqrt{\frac{\ell_{Ci}^2}{\ell_{Ci}^2 + 2}}\right)\mathbb{E}\left[\Delta_{A|C}(X_2)\Delta_{A|C}(X_2')\right]\,\mathbb{E}\left[\Delta_{B|C}(X_3)\Delta_{B|C}(X_3')\right]$$

# G   Type-I Bound Proofs

## G.1   Moment-matched normal against a threshold

**Theorem 6.2.** *Assume that $A \perp\!\!\!\perp B \mid C$. Let $Z_1 = \widehat{\text{KCI}}_n$ and $Z_2 \sim \mathcal{N}\left(\widehat{\text{KCI}}, \text{Var}(\widehat{\text{KCI}}_n)\right)$ be a normal variable moment-matched to $Z_1$. Let $q > 0$ and $\rho \in (0, 1)$; define $T_1 = \sqrt{(1-\rho)/\rho}$ and $T_2 = \Phi^{-1}(1 - \rho)$, where $\Phi$ is the standard normal CDF. Then the following holds for $i \in \{1, 2\}$:*

$$\Pr\left(Z_i > \frac{q}{n}\right) \leq \rho \quad \text{if } q \geq n\widehat{\text{KCI}} + T_i\,\sqrt{n^2\,\text{Var}(\widehat{\text{KCI}}_n)}.$$

*Proof.* Notice that, when either bound is satisfied, we have that

$$\left(\frac{q}{n} - \widehat{\mathrm{KCI}}\right) \Big/ \sqrt{\mathrm{Var}(\widehat{\mathrm{KCI}}_n)} \geq T_i.$$

The result for $\widehat{\mathrm{KCI}}_n$ follows by Cantelli's inequality, which slightly improves the better-known Chebyshev inequality for one-sided bounds; it says that for any random variable $X$,

$$\Pr(X \geq \mathbb{E}[X] + \lambda) \leq \frac{\mathrm{Var}(X)}{\mathrm{Var}(X) + \lambda^2}, \quad \forall \lambda > 0,$$

and so, equivalently,

$$\Pr\left(\frac{X - \mathbb{E}\, X}{\sqrt{\mathrm{Var}(X)}} \geq t\right) \leq \frac{\mathrm{Var}(X)}{\mathrm{Var}(X) + t^2\, \mathrm{Var}(X)} = \frac{1}{1 + t^2}.$$

Plugging in $T_1$ yields that

$$\Pr\left(\widehat{\mathrm{KCI}}_n \geq \frac{q}{n}\right) \leq \Pr\left(\frac{\widehat{\mathrm{KCI}}_n - \widehat{\mathrm{KCI}}}{\sqrt{\mathrm{Var}(\widehat{\mathrm{KCI}}_n)}} > \sqrt{\frac{1 - \rho}{\rho}}\right) \leq \frac{1}{1 + \frac{1-\rho}{\rho}} = \rho,$$

as desired. The bound for $Z_2$ is similar:

$$\Pr\left(Z_2 \geq \frac{q}{n}\right) \leq \Pr\left(\frac{Z_2 - \mathbb{E}\, Z_2}{\sqrt{\mathrm{Var}(Z_2)}} \geq \Phi^{-1}(1 - \rho)\right) = 1 - \Phi\left(\Phi^{-1}(1 - \rho)\right) = \rho. \qquad \square$$

### G.2 Alignment to wild bootstrap

We provide a bound on the distance between two null distributions used in testing:

1. Wild bootstrap distribution given the test dataset.

2. Normal approximation to the test statistic $n\widehat{\mathrm{KCI}}_n$ when regression errors are present.

**Setup.** Let $\widehat{H} \in \mathbb{R}^{n \times n}$ be the kernel matrix with noisy regression under the null hypothesis, with entries $\widehat{H}_{ij} = \hat{h}_{ij}$. We define a random variable

$$Y := \frac{1}{n} \sum_{1 \leq i \neq j \leq n} \hat{h}_{ij}\, \varepsilon_i\, \varepsilon_j,$$

where $\{\varepsilon_i\}_{i=1}^n$ are i.i.d. $\mathcal{N}(0, 1)$ variables. It is known from the results of Imhof (1961) that the wild bootstrap distribution of $Y \mid \widehat{H}$ is the same as

$$(Y \mid \widehat{H}) \equiv Q := \sum_{r=1}^n \lambda_r\, (X_r^2 - 1),$$

where $X_r \sim \mathcal{N}(0, 1)$ i.i.d., and $\{\lambda_r\}_{r=1}^n$ are the eigenvalues of $\widehat{H}/n$. This centered form $(X_r^2 - 1)$ ensures that $\mathbb{E}[Q] = 0$. The variance is $\mathrm{Var}(Q) = 2 \sum_{r=1}^n \lambda_r^2 = \frac{2}{n^2} \mathrm{tr}(\widehat{H}^2)$. And the third central moment of $Q$ is $8 \sum_{r=1}^n \lambda_r^3 = \frac{8}{n^3} \mathrm{tr}(\widehat{H}^3)$ (see Buckley and Eagleson, 1988). Moreover, in the limit $n \to \infty$, $Q$ and $n\widehat{\mathrm{KCI}}_n$ under a "perfect regression" null converge to the same distribution (see Leucht and Neumann 2013, Theorem 2.1 and Pogodin et al. 2024, Theorem 4).

When regression errors are present, the errors include a small but nonzero leading variance term, and thus the null distribution of $\widehat{\mathrm{KCI}}_n$ becomes slightly *non-degenerate*. By a suitable central limit theorem argument (analogous to Theorem 5.1), $\widehat{\mathrm{KCI}}_n$ is approximately normal for large $n$:

$$\frac{(\widehat{\mathrm{KCI}}_n - \widehat{\mathrm{KCI}})}{\sqrt{\mathrm{Var}(\widehat{\mathrm{KCI}}_n)}} \xrightarrow{d} \mathcal{N}(0, 1).$$

Recall that for a second-order U-statistic with kernel $\hat{h}_{ij} = \hat{h}(X_i, X_j)$, a standard formula (Serfling, 1980, Chapter 5.2) gives

$$\mathrm{Var}(\widehat{\mathrm{KCI}}_n) = \frac{4(n-2)}{n(n-1)} \underset{i}{\mathrm{Var}}\big[\underset{j}{\mathbb{E}}(\hat{h}_{ij})\big] + \frac{2}{n(n-1)} \mathrm{Var}\big[\hat{h}_{ij}\big]$$

$$= \frac{4(n-2)}{n(n-1)} \underset{i}{\mathrm{Var}}\big[\underset{j}{\mathbb{E}}(\hat{h}_{ij})\big] + \frac{2}{n(n-1)} \mathbb{E}\big[\hat{h}_{ij}^2\big] - \frac{2}{n(n-1)} \big(\mathbb{E}[\hat{h}_{ij}]\big)^2.$$

Meanwhile, for the wild-bootstrap statistic, we have

$$\mathrm{Var}(Y \mid \widehat{H}) = \mathrm{Var}(Q) = 2 \sum_{r=1}^{n} \lambda_r^2 = \frac{2}{n^2} \mathrm{tr}(\widehat{H}^2) \longrightarrow 2\,\mathbb{E}\big[\hat{h}_{ij}^2\big] \quad \text{as } n \to \infty.$$

If the test uses correct regressions, under the null, $\mathbb{E}[h_{ij}] = \mathrm{Var}_i\big[\mathbb{E}_j(h_{ij})\big] = 0$, then $\mathrm{Var}(Q)$ converges exactly to $n^2\,\mathrm{Var}(\widehat{\mathrm{KCI}}_n)$. In the presence of regression errors, however, note in $n^2\,\mathrm{Var}(\widehat{\mathrm{KCI}}_n)$ that the factor $4\,n\,\mathrm{Var}_i[\mathbb{E}_j(\hat{h}_{ij})]$ can remain substantial if $\mathrm{Var}_i[\mathbb{E}_j(\hat{h}_{ij})]$ does not vanish. This $\mathrm{Var}_i[\mathbb{E}_j(\hat{h}_{ij})]$ term can contribute a larger leading order when multiplied by $n$. Hence, if $\mathrm{Var}_i[\mathbb{E}_j(\hat{h}_{ij})]$ is non-negligible, for large $n$, $n^2\,\mathrm{Var}(\widehat{\mathrm{KCI}}_n)$ can be bigger than $\mathrm{Var}(Q)$.

In practice, we use wild bootstrap to sample from the distribution of $Y \mid \widehat{H}$ under the noisy-regression null to determine a test threshold. Meanwhile, the actual test statistic $n\,\widehat{\mathrm{KCI}}_n$ can be approximated by a normal variable

$$S \equiv n\,Z_n \sim \mathcal{N}\left(n\widehat{\mathrm{KCI}}, n^2\,\mathrm{Var}(\widehat{\mathrm{KCI}}_n)\right).$$

Hence, we want to quantify the distance between the distribution of $Y \mid \widehat{H}$ and the distribution of $nZ_n$. Concretely, we measure

$$\sup_{x \in \mathbb{R}} \Big| \mathrm{Pr}\big(Y \mid \widehat{H} \le x\big) - \mathrm{Pr}\big(n\,Z_n \le x\big) \Big|. \tag{12}$$

A small supremum indicates that $Y \mid \widehat{H}$ (wild bootstrap) and $nZ_n$ (normal approximation under regression error) produce nearly identical thresholds, while a large value implies a more significant discrepancy between the two distributions.

**Theorem 6.3.** *Assume $A \perp\!\!\!\perp B \mid C$, and let $Y = \frac{1}{n} \sum_{i,j=1}^{n} \hat{h}_{ij}\,\varepsilon_i\,\varepsilon_j$, where $\varepsilon_i \overset{\mathrm{iid}}{\sim} \mathcal{N}(0,1)$. Let $\widehat{H}$ be the matrix with entries $\hat{h}_{i,j}$; assume $\delta := \|\widehat{H}\|_\infty^2 / \|\widehat{H}\|_2^2 < 1/2$. Let $Z_n \sim \mathcal{N}\big(\widehat{\mathrm{KCI}}, \mathrm{Var}(\widehat{\mathrm{KCI}}_n)\big)$. Define the standardized mean shift $b_{\widehat{\mathrm{KCI}}} = \frac{\widehat{\mathrm{KCI}}}{\sqrt{\mathrm{Var}(\widehat{\mathrm{KCI}}_n)}}$ and variance mismatch $\kappa_{\mathrm{var}} = \frac{\mathrm{Var}(Y|\widehat{H})}{n^2\,\mathrm{Var}(\widehat{\mathrm{KCI}}_n)}$. Further define $R_1 = \frac{2\pi^2}{3}\,\mathrm{Skew}(Y \mid \widehat{H}) = \frac{4}{3}\sqrt{2}\,\pi^2 \|\widehat{H}\|_3^3 / \|\widehat{H}\|_2^3$, $R_2 = \frac{2^{-5/4}\,\pi^{-2}}{\sqrt{1-2\,\delta}}$, and $R_3 = (2\pi)^{-7/2}$. Then, for $\Psi(x) = \frac{1}{\sqrt{2\pi}}\,x\,\exp(2\pi x)$,*

$$\sup_{x \in \mathbb{R}} \Big| \mathrm{Pr}\big(Y \mid \widehat{H} \le x\big) - \mathrm{Pr}\big(n\,Z_n \le x\big) \Big| \le \Psi\left( R_1 \kappa_{\mathrm{var}}^{3/2} + b_{\widehat{\mathrm{KCI}}} + \pi|\kappa_{\mathrm{var}} - 1| \right) + R_2 \kappa_{\mathrm{var}}^{-1/2} + R_3,$$

*Proof.* The overarching goal is to bound $\sup_{x \in \mathbb{R}} \big| \mathrm{Pr}(Q \le x) - \mathrm{Pr}(S \le x) \big|$, where $Q$ is a centered weighted sum of chi-squared variables (which, as noted above, is exactly the distribution of $Y \mid H$), and $S \equiv nZ_n$ is a normal approximation to a U-statistic-based test statistic. The classical approach (Buckley and Eagleson, 1988; J.-T. Zhang, 2005) utilizes characteristic functions ($\psi(\cdot)$) and the Fourier inversion formula to control the Kolmogorov distance between distributions.

Let $T$ be a generic random variable with characteristic function $\psi_T(t) = \mathbb{E}[e^{itT}]$. If $\log\big(\psi_T(t)\big)$ admits the power series expansion

$$\log(\psi_T(t)) = \sum_{l=1}^{\infty} \mathcal{K}_l(T)\,\frac{(it)^l}{l!},$$

then the constants $\mathcal{K}_\ell(T)$ for $\ell = 1, 2, \dots$ ) are the cumulants of $T$ (Muirhead, 2009, Sec. 2.4). In particular: $\mathcal{K}_1(T) = \mathbb{E}[T]$ is the mean, $\mathcal{K}_2(T) = \mathrm{Var}(T)$ is the variance, $\mathcal{K}_3(T) = \mathbb{E}[(T - \mathbb{E}[T])^3]$ is the third cumulants, with $\mathcal{K}_3(T)/\mathcal{K}_2(T)^{3/2} = \mathrm{Skew}(T)$ being the skewness.

Recall that $Q = \sum_{r=1}^{n} \lambda_r \left( Z_r^2 - 1 \right)$, where $Z_r \sim \mathcal{N}(0,1)$ i.i.d., and $\{\lambda_r\}$ are positive (eigenvalues of $H/n$). By construction,

$$\mathcal{K}_1(Q) = 0, \quad \mathcal{K}_2(Q) = 2 \sum_{r=1}^{n} \lambda_r^2, \quad \mathcal{K}_3(Q) = 8 \sum_{r=1}^{n} \lambda_r^3, \quad \mathcal{K}_l(Q) = 2^{l-1} (l-1)! \sum_{r=1}^{n} \lambda_r^l \quad (l \geq 3).$$

Since $\{\lambda_r\}_{r=1}^{n}$ are the eigenvalues of the Hermitian matrix $\widehat{H}/n$, the cumulants can be expressed in terms of the Schatten norms of $\widehat{H}/n$. For simplicity, define $\widetilde{H} := \widehat{H}/n$. The Schatten p-norm of $\widetilde{H}$ is

$$\|\widetilde{H}\|_p = \left( \sum_r^n \lambda_r^p \right)^{\frac{1}{p}}, \text{ if } p \leq \infty, \text{ and } \quad \|\widetilde{H}\|_\infty = \max_{1 \leq r \leq n} \lambda_r$$

In particular, $\mathcal{K}_2(Q) = 2\|\widetilde{H}\|_2^2$, and $\mathcal{K}_3(Q) = 8\|\widetilde{H}\|_3^3$, which provides a more interpretable way to quantify $Q$'s variance and skewness based on the spectrum of $\widetilde{H}$.

Define the normalized version $Q^*$ by

$$Q^* = \frac{Q - \mathbb{E}[Q]}{\sqrt{\mathrm{Var}(Q)}} = \frac{Q}{\sqrt{\mathcal{K}_2(Q)}}.$$

Hence, $\mathcal{K}_1(Q^*) = 0, \mathcal{K}_2(Q^*) = 1, \mathcal{K}_3(Q^*) = 8 \sum_{r=1}^{n} \lambda_r^3 / \mathcal{K}_2^{3/2}(Q)$, and for $l \geq 3$,

$$\mathcal{K}_\ell(Q^*) = \frac{\mathcal{K}_\ell(Q)}{(\mathcal{K}_2(Q))^{\ell/2}}.$$

For ease of comparison with Q, define

$$S^* = \frac{S}{\sqrt{\mathcal{K}_2(Q)}} \sim \mathcal{N}\left( \frac{n \widehat{\mathrm{KCI}}}{\sqrt{\mathcal{K}_2(Q)}}, \frac{n^2 \, \mathrm{Var}(\widehat{\mathrm{KCI}}_n)}{\mathcal{K}_2(Q)} \right).$$

The distance $\sup_x |\Pr(Q \leq x) - \Pr(S \leq x)|$ is equivalent to comparing $Q^*$ and $S^*$: $\sup_{x \in \mathbb{R}} |\Pr(Q^* \leq x) - \Pr(S^* \leq x)|$.

By results of Esseen (1945, page 33), we have

$$\sup_{x \in \mathbb{R}} |\Pr(Q^* \leq x) - \Pr(S^* \leq x)| \leq \frac{1}{2\pi} \int_{-\infty}^{+\infty} \left| \frac{\psi_{Q^*}(t) - \psi_{S^*}(t)}{t} \right| dt,$$

where $\psi_{Q^*}$ and $\psi_{S^*}$ are the characteristic functions of $Q^*$ and $S^*$, respectively.

$$\psi_{Q^*}(t) = \prod_{r=1}^{n} \exp\left( -it \frac{\lambda_r}{\mathcal{K}_2^{1/2}(Q)} \right) \cdot \left( 1 - it \frac{2\lambda_r}{\mathcal{K}_2^{1/2}(Q)} \right)^{-1/2}$$

$$\psi_{S^*}(t) = \exp\left( it \frac{n\widehat{\mathrm{KCI}}}{\mathcal{K}_2^{1/2}(Q)} - t^2 \frac{n^2 \, \mathrm{Var}(\widehat{\mathrm{KCI}}_n)}{2\mathcal{K}_2(Q)} \right).$$

To handle the integral

$$\int_{-\infty}^{\infty} \left| \frac{\psi_{Q^*}(t) - \psi_{S^*}(t)}{t} \right| dt,$$

it is standard to split the domain at $|t| = A$ for some positive $A$. Define

$$I_1 = \int_{|t| \leq A} \left| \frac{\psi_{Q^*}(t) - \psi_{S^*}(t)}{t} \right| dt, \quad I_2 = \int_{|t| > A} \left| \frac{\psi_{S^*}(t)}{t} \right| dt, \quad I_3 = \int_{|t| > A} \left| \frac{\psi_{Q^*}(t)}{t} \right| dt.$$

Then,

$$\int_{-\infty}^{\infty} \left| \frac{\psi_{Q^*}(t) - \psi_{S^*}(t)}{t} \right| dt \leq I_1 + I_2 + I_3.$$

Optimizing over $A$ balances these different regions. This is a classical technique in Fourier-based proofs of Berry–Esseen-type inequalities.

**Bounding $I_1$.** We decompose $I_1$ based on the characteristic function ratio. Define
$$r(t) := \log\big(\psi_{Q^*}(t)\big) \,-\, \log\big(\psi_{S^*}(t)\big).$$
Then
$$
I_1 = \int_{|t|\leq A} |\psi_{S^*}(t)| \left| \frac{\psi_{Q^*}(t)/\psi_{S^*}(t) - 1}{t} \right| dt
$$
$$
= \int_{|t|\leq A} |\psi_{S^*}(t)| \left| \frac{\exp(r(t)) - 1}{t} \right| dt
$$
$$
\leq \int_{|t|\leq A} |\psi_{S^*}(t)| \frac{|r(t)|\exp(|r(t)|)}{|t|} dt,
$$
where the last step comes from the inequality $|\exp(z) - 1| \leq |z|\exp(|z|)$.

We use the following expansion bound for real $\theta$, which be easily verified using the mean-value theorem (see also Buckley and Eagleson, 1988; J.-T. Zhang, 2005):
$$
\left| \log(1 + i\theta) - \left\{ i\theta + \frac{\theta^2}{2} \right\} \right| \leq |\theta|^3/3. \tag{13}
$$

Concretely,
$$
r(t) = \left( i\,t\,\frac{\sum_{r=1}^n \lambda_r}{\mathcal{K}_2(Q)^{1/2}} \right) + \frac{1}{2}\sum_{r=1}^n \log\!\left( 1 - i\,t\,\frac{2\,\lambda_r}{\mathcal{K}_2(Q)^{1/2}} \right) \;+\; \left( i\,t\,\frac{n\,\widehat{\mathrm{KCI}}}{\mathcal{K}_2^{1/2}(Q)} - t^2\,\frac{n^2\,\mathrm{Var}(\widehat{\mathrm{KCI}}_n)}{2\,\mathcal{K}_2(Q)} \right).
$$

By bounding each $\log(\cdot)$ via the expansion (13), we obtain
$$
|r(t)| \;\leq\; \frac{1}{6}\frac{\sum_{r=1}^n |2\,t\,\lambda_r|^3}{\mathcal{K}_2^{3/2}(Q)} \;+\; \left| i\,t\,\frac{n\,\widehat{\mathrm{KCI}}}{\mathcal{K}_2^{1/2}(Q)} \right| \;+\; \left| t^2 \left( \frac{\sum_{r=1}^n \lambda_r^2}{\mathcal{K}_2(Q)} - \frac{n^2\,\mathrm{Var}(\widehat{\mathrm{KCI}}_n)}{2\,\mathcal{K}_2(Q)} \right) \right|.
$$
Recognizing $\mathcal{K}_3(Q^*) = 8\sum_r \lambda_r^3/(\mathcal{K}_2(Q))^{3/2}$ and $\sum_r \lambda_r^2 = \frac{1}{2}\mathcal{K}_2(Q)$, we rewrite:
$$
|r(t)| \;\leq\; \frac{1}{6}\,|t|^3\,\mathcal{K}_3(Q^*) \;+\; |t|\,\frac{n\,\widehat{\mathrm{KCI}}}{\mathcal{K}_2^{1/2}(Q)} \;+\; \frac{t^2}{2}\left| \frac{n^2\,\mathrm{Var}(\widehat{\mathrm{KCI}}_n)}{\mathcal{K}_2(Q)} - 1 \right|. \tag{14}
$$

Hence, for $|t|\leq A$,
$$
I_1 \;\leq\; \exp\big(|r(A)|\big)\int_{|t|\leq A} \exp\!\left( -\frac{t^2\,n^2\,\mathrm{Var}(\widehat{\mathrm{KCI}}_n)}{2\,\mathcal{K}_2(Q)} \right)\left( \frac{1}{6}\,t^2\,\mathcal{K}_3(Q^*) \right.
$$
$$
\left. +\; \frac{n\,\widehat{\mathrm{KCI}}}{\mathcal{K}_2(Q)^{1/2}} \;+\; \frac{|t|}{2}\left| \frac{n^2\,\mathrm{Var}(\widehat{\mathrm{KCI}}_n)}{\mathcal{K}_2(Q)} - 1 \right| \right) dt.
$$

This splits naturally into three integrals:
$$
I_1 \;\leq\; \frac{1}{6}\,\exp\big(|r(A)|\big)\,\mathcal{K}_3(Q^*)\int_{|t|\leq A} t^2\,\exp\!\left( -\frac{t^2\,n^2\,\mathrm{Var}(\widehat{\mathrm{KCI}}_n)}{2\,\mathcal{K}_2(Q)} \right) dt
$$
$$
+\; \frac{1}{2}\,\exp\big(|r(A)|\big)\left| \frac{n^2\,\mathrm{Var}(\widehat{\mathrm{KCI}}_n)}{\mathcal{K}_2(Q)} - 1 \right|\int_{|t|\leq A} |t|\,\exp\!\left( -\frac{t^2\,n^2\,\mathrm{Var}(\widehat{\mathrm{KCI}}_n)}{2\,\mathcal{K}_2(Q)} \right) dt
$$
$$
+\; \exp\big(|r(A)|\big)\,\frac{n\,\widehat{\mathrm{KCI}}}{\mathcal{K}_2(Q)^{1/2}}\int_{|t|\leq A} \exp\!\left( -\frac{t^2\,n^2\,\mathrm{Var}(\widehat{\mathrm{KCI}}_n)}{2\,\mathcal{K}_2(Q)} \right) dt.
$$

In each term, the integral is bounded by Gaussian-like tail and one can get explicit numerical constants:
$$
I_1 \leq \frac{1}{3}\sqrt{\frac{\pi}{2}}\,\exp\big(|r(A)|\big)\,\mathcal{K}_3(Q^*)\left( \frac{\mathcal{K}_2(Q)}{n^2\,\mathrm{Var}(\widehat{\mathrm{KCI}}_n)} \right)^{3/2} + \exp\big(|r(A)|\big)\left| \frac{\mathcal{K}_2(Q)}{n^2\,\mathrm{Var}(\widehat{\mathrm{KCI}}_n)} - 1 \right|
$$
$$
+\; \sqrt{2\pi}\,\exp\big(|r(A)|\big)\,\frac{\widehat{\mathrm{KCI}}}{\sqrt{\mathrm{Var}(\widehat{\mathrm{KCI}}_n)}} \tag{15}
$$

**Boudning $I_2$.** Recall

$$I_2 = \int_{|t|>A} \left| \frac{\psi_{S^*}(t)}{t} \right| dt,$$

where

$$\psi_{S^*}(t) = \exp\left( i\,t\,\frac{n\,\widehat{\mathrm{KCI}}}{\mathcal{K}_2(Q)^{1/2}} - \frac{t^2\,n^2\,\mathrm{Var}(\widehat{\mathrm{KCI}}_n)}{2\,\mathcal{K}_2(Q)} \right).$$

Since $\left| \psi_{S^*}(t) \right| = \exp\left( -\frac{t^2\,n^2\,\mathrm{Var}(\widehat{\mathrm{KCI}}_n)}{2\,\mathcal{K}_2(Q)} \right)$, we have

$$I_2 = \int_{|t|>A} \frac{1}{|t|}\, \exp\left( -\frac{t^2\,n^2\,\mathrm{Var}(\widehat{\mathrm{KCI}}_n)}{2\,\mathcal{K}_2(Q)} \right) dt.$$

Next, use the fact that $|t| \geq A$ implies $\frac{1}{|t|} \leq \frac{t^2}{A^3}$. Hence,

$$I_2 \leq \int_{|t|>A} \frac{t^2}{A^3}\, \exp\left( -\frac{t^2\,n^2\,\mathrm{Var}(\widehat{\mathrm{KCI}}_n)}{2\,\mathcal{K}_2(Q)} \right) dt.$$

$$\leq \frac{\sqrt{2\pi}}{A^3} \left( \frac{\mathcal{K}_2(Q)}{n^2\,\mathrm{Var}(\widehat{\mathrm{KCI}}_n)} \right)^{3/2} \tag{16}$$

**Bounding $I_3$.** Recall

$$I_3 = \int_{|t|>A} \left| \frac{\psi_{Q^*}(t)}{t} \right| dt.$$

Following J.-T. Zhang (2005, (B.3)), we have

$$|\psi_{Q^*}(t)| \leq \left( \sum_{k=1}^{n} \tau_k (2t^2)^k \right)^{-\frac{1}{4}},$$

where $\tau_k = \sum_{j_1 < j_2 < \cdots < j_k} \alpha_{j_1} \alpha_{j_2} \dots \alpha_{j_k}$ and $\alpha_r = \frac{2\lambda_r^2}{\mathcal{K}_2(Q)} = \frac{\lambda_r^2}{\sum_{j=1}^{n} \lambda_j^2}$. Let $\delta = \max_{1 \leq r \leq n} \alpha_r$. Using induction on $k$ we can prove that,

$$\tau_k \geq \frac{1}{k!}(1 - k\delta)^k.$$

When $\delta < 1/2$, $n$ is at least 2, such that

$$|\psi_{Q^*}(t)| \leq \tau_2^{-1/4}(2t^2)^{-1/2} \leq 2^{1/4}(1 - 2\delta)^{-1/2}(2t^2)^{-1/2}.$$

Recall that $\|\widetilde{H}\|_2 = \left( \sum_r^n \lambda_r^2 \right)^{\frac{1}{2}}$ and $\|\widetilde{H}\|_\infty = \max_{1 \leq r \leq n} \lambda_r$. If $\delta < 1/2$, then the largest eigenvalue contributes less than half of the total squared eigenvalue mass. Since $\delta = \|\widetilde{H}\|_\infty^2 / \|\widetilde{H}\|_2^2 = \|\widehat{H}\|_\infty^2 / \|\widehat{H}\|_2^2$, $\delta < 1/2$ implies $\|\widehat{H}\|_2^2 > 2\|\widehat{H}\|_\infty^2$.

Assume $\delta < 1/2$, it follows that

$$I_3 = \int_{|t|>A} \left| \frac{\psi_{Q^*}(t)}{t} \right| dt \leq 2^{3/4}(1 - 2\delta)^{-1/2} \int_{t \geq A} t^{-2}\, dt = 2^{3/4}(1 - 2\delta)^{-1/2}A^{-1}. \tag{17}$$

**Combing $I_1$, $I_2$ and $I_3$.** From (15), (16), and (17), we obtain

$$I_1 + I_2 + I_3 \leq \frac{2^{3/4}}{\sqrt{1 - 2\delta}A} + \sqrt{2\pi}\exp\left(|r(A)|\right)\frac{\widehat{\mathrm{KCI}}}{\sqrt{\mathrm{Var}(\widehat{\mathrm{KCI}}_n)}} + \exp\left(|r(A)|\right)\left| \frac{\mathcal{K}_2(Q)}{n^2\,\mathrm{Var}(\widehat{\mathrm{KCI}}_n)} - 1 \right|$$

$$+ \left( \frac{1}{3}\sqrt{\frac{\pi}{2}}\exp\left(|r(A)|\right)\mathcal{K}_3(Q^*) + \frac{\sqrt{2\pi}}{A^3} \right)\left( \frac{\mathcal{K}_2(Q)}{n^2\,\mathrm{Var}(\widehat{\mathrm{KCI}}_n)} \right)^{3/2}. \tag{18}$$

Here $A > 0$ is a splitting parameter, which we leave unspecified but may be chosen arbitrarily, and the quantity $|r(A)|$ is bounded as in (14).

**Final Assembly.** Combining the pieces, we have

$$\int_{-\infty}^{\infty} \left| \frac{\psi_{Q^*}(t) - \psi_{S^*}(t)}{t} \right| dt \ \leq \ (I_1 + I_2 + I_3)$$

which in turn implies the Kolmogorov bound

$$\sup_{x \in \mathbb{R}} \left| \Pr(Q^* \leq x) - \Pr(S^* \leq x) \right| \ \leq \ \frac{1}{2\pi} (I_1 + I_2 + I_3).$$

Putting everything together, the distributional distance satisfies:

$$\sup_{x \in \mathbb{R}} \left| \Pr(Q \leq x) \ - \ \Pr(S \leq x) \right|$$

$$\leq \ \frac{1}{2^{1/4} \pi \sqrt{(1 - 2\delta)A}} + \left( \frac{\exp(|r(A)|) \, \mathcal{K}_3(Q^*)}{6\sqrt{2\pi}} + \frac{1}{\sqrt{2\pi}A^3} \right) \left( \frac{\mathcal{K}_2(Q)}{n^2 \, \mathrm{Var}(\widehat{\mathrm{KCI}}_n)} \right)^{3/2}$$

$$+ \frac{\exp(|r(A)|)}{\sqrt{2\pi}} \frac{\widehat{\mathrm{KCI}}}{\sqrt{\mathrm{Var}(\widehat{\mathrm{KCI}}_n)}} + \frac{\exp(|r(A)|)}{2\pi} \left| \frac{\mathcal{K}_2(Q)}{n^2 \, \mathrm{Var}(\widehat{\mathrm{KCI}}_n)} - 1 \right|.$$

Here $\delta = \|\widehat{H}\|_\infty^2 / \|\widehat{H}\|_2^2$, and the parameter $A > 0$ is arbitrary. The exponential factor $\exp(|r(A)|)$ is controlled by

$$\exp(|r(A)|) \ \leq \ \exp\left( \frac{1}{6}A^3 \mathcal{K}_3(Q^*) \ + \ A \frac{n \, \widehat{\mathrm{KCI}}}{\mathcal{K}_2^{1/2}(Q)} \ + \ \frac{A^2}{2} \left| \frac{n^2 \, \mathrm{Var}(\widehat{\mathrm{KCI}}_n)}{\mathcal{K}_2(Q)} - 1 \right| \right).$$

Recall that $\mathcal{K}_2(Q) = \mathrm{Var}(Q) = \mathrm{Var}(Y \mid \widehat{H})$ and $\mathcal{K}_3(Q^*) = \mathrm{Skew}(Q) = \mathrm{Skew}(Y \mid \widehat{H})$. Define the standardized mean shift and variance mismatch scale by

$$b_{\widehat{\mathrm{KCI}}} := \frac{\widehat{\mathrm{KCI}}}{\sqrt{\mathrm{Var}(\widehat{\mathrm{KCI}}_n)}}, \qquad \kappa_{\mathrm{var}} := \frac{\mathrm{Var}(Y \mid \widehat{H})}{n^2 \, \mathrm{Var}(\widehat{\mathrm{KCI}}_n)}$$

We now fix $A = 2\pi\sqrt{\kappa_{\mathrm{var}}} = 2\pi\sqrt{\frac{\mathrm{Var}(Y|\widehat{H})}{n^2 \, \mathrm{Var}(\widehat{\mathrm{KCI}}_n)}}$. With this choice,

$$\exp(|r(\sqrt{\kappa_{\mathrm{var}}})|) \ \leq \ \exp\left( 2\pi \left( \frac{2\pi^2}{3} \mathrm{Skew}(Y \mid \widehat{H})\kappa_{\mathrm{var}}^{3/2} \ + \ b_{\widehat{\mathrm{KCI}}} \ + \ \pi|\kappa_{\mathrm{var}} - 1| \right) \right).$$

Consequently, we obtain

$$\sup_{x \in \mathbb{R}} \left| \Pr(Y \mid \widehat{H} \leq x) - \Pr(n\, Z_n \leq x) \right|$$

$$\leq \ \frac{\exp(|r(\sqrt{\kappa_{\mathrm{var}}})|)}{\sqrt{2\pi}} \left( \frac{1}{6} \mathrm{Skew}(Y \mid \widehat{H})\kappa_{\mathrm{var}}^{3/2} + b_{\widehat{\mathrm{KCI}}} + \frac{1}{\sqrt{2\pi}} \left| \kappa_{\mathrm{var}} - 1 \right| \right)$$

$$+ \frac{\kappa_{\mathrm{var}}^{-1/2}}{2^{5/4} \pi^2 \sqrt{(1 - 2\delta)}} + \frac{1}{(2\pi)^{7/2}}$$

Finally, introducing the shorthand $\Psi(x) := \frac{1}{\sqrt{2\pi}} x \exp(2\pi x)$, the bound can be written compactly as

$$\sup_{x \in \mathbb{R}} \left| \Pr(Y \mid \widehat{H} \leq x) - \Pr(n\, Z_n \leq x) \right|$$

$$\leq \ \Psi\left( \frac{2\pi^2}{3} \mathrm{Skew}(Y \mid \widehat{H})\kappa_{\mathrm{var}}^{3/2} + b_{\widehat{\mathrm{KCI}}} + \pi|\kappa_{\mathrm{var}} - 1| \right) + \frac{\kappa_{\mathrm{var}}^{-1/2}}{2^{5/4} \pi^2 \sqrt{(1 - 2\delta)}} + \frac{1}{(2\pi)^{7/2}}. \quad (19)$$

**Interpretation of the terms in** (19)**.**

- **Perfect regression.** If the conditional mean embeddings are estimated without error, then $b_{\widehat{\mathrm{KCI}}} = 0$ and $\kappa_{\mathrm{var}} = 1$. In this regime, the bound reflects the intrinsic discrepancy between the wild bootstrap distribution and its Gaussian approximation.

- **Imperfect regression.** When regression is imperfect, the bound is inflated by two effects: (i) the mean-shift term $b_{\widehat{\mathrm{KCI}}}$, which quantifies the bias introduced by regression errors in the KCI statistic, and (ii) the variance-mismatch term $\kappa_{\mathrm{var}}$, which captures deviations between the empirical variance and its wild bootstrap approximation. Both effects arise from noise in the conditional mean embedding estimates and directly degrade the quality of the wild bootstrap null calibration.

Finally, note that $\mathrm{Var}(Y \mid \widehat{H})$ is the variance of a weighted sum of chi-squared random variables, and $\mathrm{Var}(Y \mid \widehat{H}) \to 2\,\mathbb{E}[\hat{h}_{ij}^2]$. Moreover, $n^2 \,\mathrm{Var}(\widehat{\mathrm{KCI}}_n) \sim 4n\nu_1 + 2\nu_2$. If regression errors are well controlled so that $\widetilde{\mathrm{KCI}} = o(1/n)$ and $\nu_1 = o(1/n)$, then $n^2 \,\mathrm{Var}(\widehat{\mathrm{KCI}}_n) \to 2\,\mathbb{E}[\hat{h}_{ij}^2]$, and consequently

$$b_{\widehat{\mathrm{KCI}}} = \frac{\widehat{\mathrm{KCI}}}{\sqrt{\mathrm{Var}(\widehat{\mathrm{KCI}}_n)}} = o(1), \qquad \kappa_{\mathrm{var}} = \frac{\mathrm{Var}(Y \mid \widehat{H})}{n^2 \,\mathrm{Var}(\widehat{\mathrm{KCI}}_n)} = 1 + o(1).$$

which ensures that the leading error terms in (19) remain controlled asymptotically.

With these expressions established, the proof is complete. $\qquad\square$

## H   Experimental Setup and Results

In this section, we include additional experimental results. Code used in our synthetic data experiments is publicly available at: `https://github.com/he-zh/kci-hardness`.

### H.1   Testing procedure

Our testing pipeline follows the general structure of KCI, with modifications inspired by SplitKCI (Pogodin et al., 2024). The data are divided into an independent training set of size $m$ and a test set of size $n$. We first select kernels $k_{C \to A}$ and $k_{C \to B}$ using leave-one-out validation via kernel ridge regression as in SplitKCI, and estimate the conditional mean embeddings $\widehat{\mu}_{A|C}$ and $\widehat{\mu}_{B|C}$ on the training data. Unlike SplitKCI, though, we do not further split the training set to obtain independent estimates $\widehat{\mu}_{A|C}^{(1)}$ and $\widehat{\mu}_{A|C}^{(2)}$; instead, both $\widehat{\mu}_{A|C}$ and $\widehat{\mu}_{B|C}$ are trained on the full training set. When evaluating on any point $c$, the conditional mean embedding for $A \mid C$ is

$$\widehat{\mu}_{A|C}(c) = \Phi_A^{\mathrm{tr}} \left( K_C^{\mathrm{tr}} + \lambda I_m \right)^{-1} K_{Cc}^{\mathrm{tr}}.$$

The training feature matrix on $A$ is $\Phi_A^{\mathrm{tr}} = [\,\phi_A(a_1), \phi_A(a_2), \ldots, \phi_A(a_m)\,]$, the training kernel matrix on $C$ is $(K_C^{\mathrm{tr}})_{i,j} = k_C(c_i, c_j)$, $\lambda$ is the ridge regression parameter, and $K_{Cc}^{\mathrm{tr}} \in \mathbb{R}^m$ is given by $(K_{Cc}^{\mathrm{tr}})_i = k_C(c_i, c)$, where $a_i$ and $c_i$ denote the training samples. The conditional mean embedding for $B \mid C$ can be computed analogously.

When the power maximization technique is applied, we additionally maximize $\widehat{\mathrm{SNR}}$ using gradient descent on the training data by selecting the kernel $k_C$. Finally, given the chosen kernels, we compute the kernel matrices $\widehat{K}_A^c$, $\widehat{K}_B^c$, and $K_C$ on the test set to estimate KCI.

Although we use the unbiased estimator in Eq.(6) for analytical convenience, in practice it is preferable to use the HSIC-like unbiased estimator (Pogodin et al., 2024, Eq.(21)):

$$\widehat{\mathrm{KCI}}_n = \frac{1}{m(m-3)} \left( \mathrm{tr}(KL) + \frac{1^\top K 1 1^\top L 1}{(m-1)(m-2)} - \frac{2}{m-1} 1^\top K L 1 \right) \qquad (20)$$

where 1 is the all-ones vector, $K = \widehat{K}_A^c$, $L = \widehat{K}_B^c \odot K_C$, and $\odot$ denotes elementwise product. This estimator "centralizes" the kernel matrices, which helps average out some regression errors. While it does not eliminate all errors, it improves the KCI estimate and stabilizes power maximization.

To obtain p-values, we approximate the null distribution using the wild bootstrap with Rademacher variables. Following SplitKCI, we generate wild bootstrap statistics

$$\widehat{V}s = \frac{1}{m(m-3)}\left(\text{tr}(\tilde{K}_s L) + \frac{1^\top \tilde{K}_s 11^\top L1}{(m-1)(m-2)} - \frac{2}{m-1}1^\top \tilde{K}_s L1\right)$$

where $\tilde{K}_s = q_s q_s^\top \odot \widehat{K}_A^c$ and $(q_s)_i = \pm 1$ independently with equal probability for all $s$ and $i$. The p-value is then computed as

$$p = \frac{1}{S}\sum_{s=1}^{S} \mathbb{1}(\widehat{\text{KCI}}_n < \widehat{V}_s).$$

## H.2 Synthetic data

**1D synthetic test case.** We compare standard KCI, KCI with power-maximizing kernel selection, and GCM, using linear kernels for $A$ and $B$ in all methods, on problem (7). Figure 4 shows that GCM, while maintaining low Type-I error, fails to detect conditional dependence in this setting. Standard KCI exhibits high Type-II error, whereas power-maximized KCI achieves low Type-II error. For both KCI variants, Type-I error decreases as the training size increases.

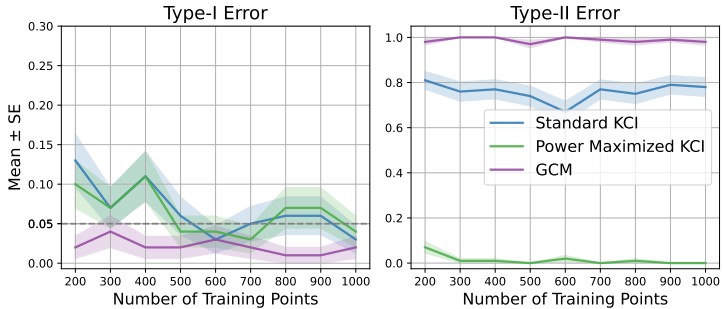

Figure 4: Mean and standard error (over 100 runs) of errors on the problem (7) with $f_A = \cos$, $f_B = \exp$, $\tau = 0.1$ and $\beta = 3$ across training sizes; the independent test set has size 200. The significance level is set at $\alpha = 0.05$

**3D synthetic test case.** We study the problems introduced in Appendix F.3. Both scenarios used $f_A = \cos$, $f_B = \exp$, noise scale $\tau = 0.1$, dependence frequency $\beta = 2$. Gaussian kernels are used for all kernels throughout. The significance level is set at $\alpha = 0.05$.

Table 1 summarizes empirical results with 200 training points for regression and 200 test points, averaged over 100 runs, with 500 training epochs. Kernel ridge regression with leave-one-out validation selects kernels for $k_{C\to A}$ and $k_{C\to B}$, with per-dimension lengthscales. Power maximization is utilized to select $k_C$ kernel.

We further compare standard KCI with fixed lengthscales and KCI with power-maximized kernel selection across training sizes 200–1000, keeping the test size fixed at 200. Results are reported for regressors trained to convergence (500 epochs) and with early stopping. Experiments are repeated 100 times, and we report mean and standard error.

We conduct experiments on two scenarios. Scenario 1: shared-coordinate dependence, where $A$ and $B$ depend on the same coordinate of $C_1$ (see Figure 5). Scenario 2: separate-coordinate dependence, where $A$ depends on $C_1$, $B$ on $C_2$, correlation on $C_3$ (see Figure 6).

In Scenario 1, Type-II error is substantially lower than in Scenario 2, as regression errors along the same coordinate are more strongly correlated, amplifying the apparent dependence. Power maximization in Scenario 1 slightly reduces Type-I error, probably because it focuses on only the coordinate $C_1$, while ignoring other two dimensions, thereby reducing noise.

In Scenario 2, Type-I error is lower overall since regression errors depend on separate coordinates and are less correlated. However, when conditional mean embeddings are undertrained, power maximization can further increase Type-I error, as it mistakenly interprets the weak dependence between regression errors as conditional dependence and amplifies it.

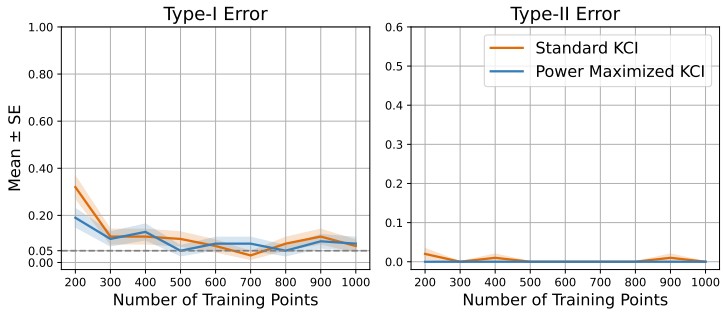

(a) Training to convergence.

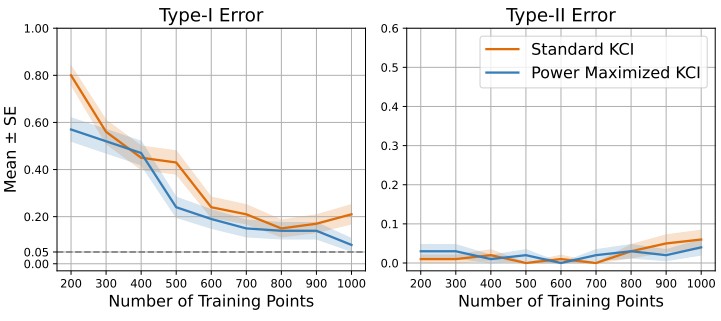

(b) Early stopping.

Figure 5: Shared-coordinate dependence. Means and standard errors (over 100 runs) of Type-I and Type-II errors on the 3D synthetic case with $f_A = \cos$, $f_B = \exp$, and $\tau = 0.1$. All kernels are Gaussian, and the significance level is set at $\alpha = 0.05$.

### H.3 Real data

We conducted experiments on the UTKFace dataset (Z. Zhang et al., 2017), following the setup of Y. Zhang et al. (2025). Although not described as such in their paper, this test is effectively a KCI test. In particular, we used the cropped and aligned UTKFace dataset to test whether age ($A$) depends on the full face image ($B$) when conditioned on the same image with a specific region masked out ($C$). The null hypothesis is $\mathbb{E}[A \mid C] = \mathbb{E}[A \mid B, C]$, corresponding to a linear kernel on $A$, where the conditional mean embedding can be estimated as a regressor from $C$ to $A$ via neural networks.

The dataset is split into ten subsets, each with its own training and test partition. We reran their code and report both the resulting $p$-values for the conditional independence tests and the mean absolute error (MAE) of the age regressors (ImageNet-pretrained) used in testing. For comparison, we also retrained the same network from scratch (random initialization) and report its MAE and corresponding test results. The results are shown in Figure 7.

In Y. Zhang et al. (2025), $p$-values remain above $5\%$ when a facial region is masked, suggesting that the region is not critical for age estimation. However, when the same network is trained from random initialization, the validation loss increases–indicating a less accurate conditional mean embedding–and the resulting $p$-values drop consistently across all regions. This shows that test outcomes are highly sensitive to regressor quality: imperfect conditional mean estimation makes the test more prone to signal dependence.

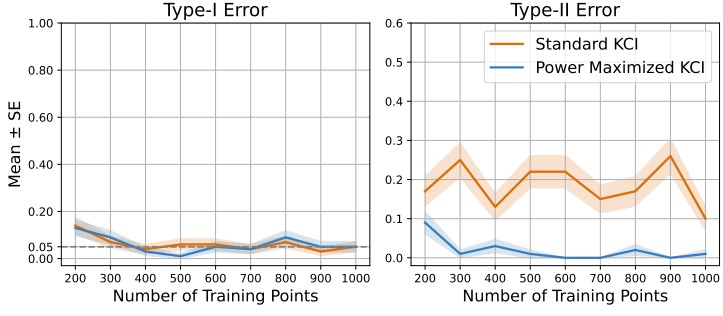

(a) Training to convergence.

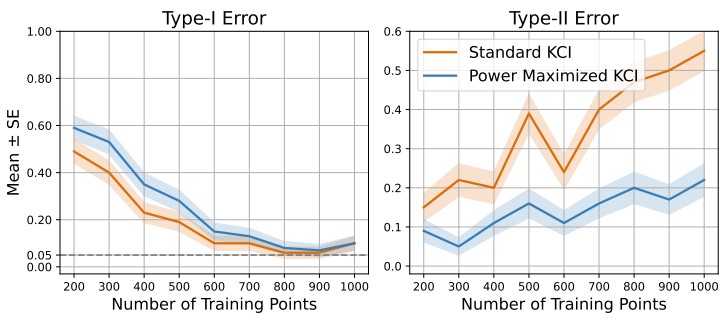

(b) Early stopping.

Figure 6: Separate-coordinate dependence. Means and standard errors (over 100 runs) of Type-I and Type-II errors on the 3D synthetic case with $f_A = \cos$, $f_B = \exp$, and $\tau = 0.1$. All kernels are Gaussian, and the significance level is set at $\alpha = 0.05$.

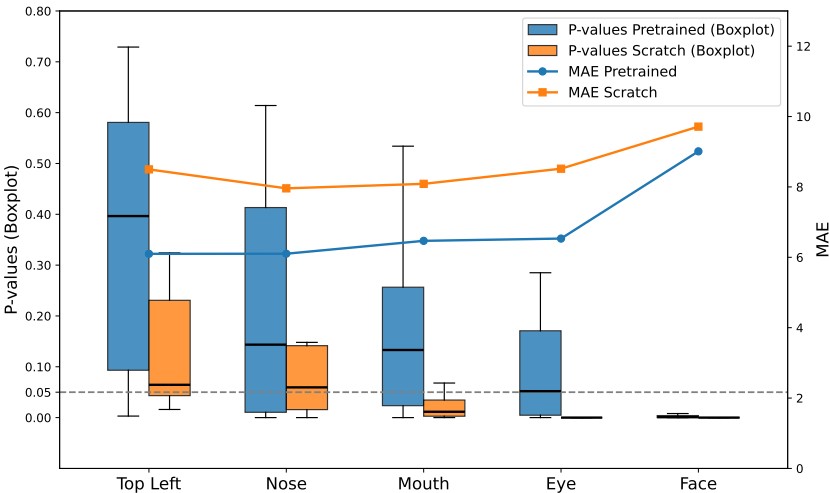

Figure 7: Box plots of $p$-values (left y-axis) and test MAE (right y-axis) across different facial regions in the age estimation task. "Pretrained" refers to using an ImageNet-pretrained age regressor, while "Scratch" indicates training the same model from random initialization.

