# OpenReview forum: "On the Hardness of Conditional Independence Testing In Practice"
_NeurIPS.cc/2025/Conference — NeurIPS 2025 spotlight_

### Official Review · Reviewer_web4 · 2025-06-27

**Clarity:** 4
**Significance:** 4
**Originality:** 3
**Rating:** 5
**Confidence:** 3

**Summary:**

This paper examines issues about conditional independence testing. It gives a good review of how this has been done over the ages and investigates kernel-based conditional independence as well as how to choose the best kernels and the associated pros and cons.

**Questions:**

In line 170, where you define $t_n$, what is $q$?

**Ethical Concerns:**

["NO or VERY MINOR ethics concerns only"]

**Limitations:**

yes

**Paper Formatting Concerns:**

None identified

**Quality:**

3

**Strengths And Weaknesses:**

This paper presents a detailed background for all the key methods used in rich statistical formulations. However the paper could benefit from performing more experiments to prove the points they are establishing.

---

> ### Author Rebuttal · Authors · 2025-07-31
>
> We thank the reviewer for the positive review and constructive suggestions.
> Below, we address the reviewer's concerns and clarify how we will revise the manuscript accordingly.
>
> ## More supporting experiments
> We thank the reviewer for this constructive suggestion. We have run substantial additional simulations beyond those in the submitted paper, which we now present to demonstrate how expirical results align with our theory.
>
>
>
> To begin with, we have additional simulation results for the settings already considered in the paper; we will discuss a new setting on images afterwards.
>
> ### 1. Setting of Section 4
>
> We compared our power maximization (PM) approach against standard KCI across training sizes 200-1000 (only parts of them are shown due to space limit), using linear kernels on $A$ and $B$, with $\beta=3$ and test size equals to 200. All experiments are repeated 100 times. We report the mean ± standard error.
> The standard KCI use fixed-bandwidth conditioning kernel while PM use learned-bandwidth conditioning kernel:
>
> |Training Size|Type I Error (standard)|Type II Error (standard)|Type I Error (PM)|Type II Error(PM)|
> |-|-|-|-|-|
> |200|0.13±0.034|0.81±0.039|0.10±0.030|0.07±0.026|
> |400|0.11±0.031|0.77±0.042|0.11±0.031|0.01±0.010|
> |600|0.03±0.017|0.67±0.047|0.04±0.020|0.02±0.014|
> |1000|0.03±0.017|0.78±0.041|0.04±0.020|0.00±0.000|
>
>
> Learning the conditioning kernel gives a huge improvement in test power, bringing Type II error from around 75% to almost 0. In this problem, it also maintains reasonable Type I error control, validating our theoretical analysis that the choice of $C$ kernel can be vital to empirical performance. On this problem, the bandwidth optimization stays in a region of reasonable Type I error, as shown in our Figure 2.
>
> ### 2. Complex 3-Dimensional Conditional Dependence Analysis (Setting of Appendix G.3)
>
> In this setting, we have a three-dimensional $C = (C_1, C_2, C_3)$, and use Gaussian kernels on $A$ and $B$, with $\beta=2$. All kernels are Gaussian; power maximization chooses per-dimension bandwidths for the $C$ kernel.
>
> Our analysis looks at two critical scenarios. However, due to space constraints, we present and discuss only one scenario here. This scenario follows the same setting as Appendix G.3 but includes more comprehensive results. The additional results will be provided in the appendix of the revised paper.
>
> #### Scenario: Multi-coordinate dependence
> Here $A$, $B$, and their conditional dependence use different coordinates, e.g. $A$'s marginal distribution depends on $C_1$, $B$'s on $C_2$, and their correlation depends on $C_3$.
>
> With sufficiently trained regressors:
> |Training Size|Type I Error (Standard)|Type II Error (Standard)|Type I Error (PM)|Type II Error (PM)|
> |-|-|-|-|-|
> |200|0.13±0.034|0.22±0.041|0.14±0.035|0.04±0.020|
> |400|0.09±0.029|0.16±0.037|0.12±0.032|0.02±0.014|
> |600|0.07±0.026|0.18±0.038|0.06±0.024|0.00±0.000|
> |800|0.06±0.024|0.14±0.035|0.03±0.017|0.00±0.000|
> |1000|0.03±0.017|0.20±0.040|0.07±0.026|0.00±0.000|
>
> With poorly trained regressors:
> |Training Size|Type I Error (Standard)|Type II Error (Standard)|Type I Error (PM)|Type II Error (PM)|
> |-|-|-|-|-|
> |200|0.49±0.050|0.15±0.036|0.59±0.049|0.09±0.029|
> |400|0.23±0.042|0.20±0.040|0.35±0.048|0.11±0.031|
> |600|0.10±0.030|0.24±0.043|0.15±0.036|0.11±0.031|
> |800|0.06±0.024|0.47±0.050|0.08±0.027|0.20±0.040|
> |1000|0.10±0.030|0.55±0.050|0.10±0.030|0.22±0.041|
>
> While the patterns are a little difficult to see in this tabular form (we'll put a figure in the revised paper), they agree well with our explanations that poorly-trained CMEs make Type I error explode compared to well-trained CMEs.
> With insufficiently-trained CMEs, power maximization makes Type I error far worse under Scenario 2.
>
> We will expand on these and related observations in the final version.
>
>
> ### 3. Real Data Experiments: UTKFace Age Prediction Study
>
> We conducted experiments using the UTKFace dataset, following the methodology from a recent paper:
>
> Y Zhang, Huang, Yang, and Shao. Testing Conditional Mean Independence Using Generative Neural Networks. ICML 2025.
>
> While Zhang et al. did not describe it this way, this test in fact is a KCI test; rather than estimating conditional mean embeddings with ridge regression, however, they use neural networks.
>
> In particular, we (following their setup) took the cropped and aligned UTKFace dataset, and ran a CI test checking whether the age ($A$) is dependent on the whole face image ($B$) when conditioned on the image with the mouth region blacked out ($C$).
>
> In their paper, p-values are above the 5% level when the mouth is covered, suggesting that this region is "not critical for age estimation".  They use ImageNet-pretrained neural networks to perform the regression task, achieving a mean absolute error on validation points of around 6 years. When we reran their code and evaluated on only 400 test samples (single run), the resulting p-value was 0.068 (above the significance level).
>
> We train the same network architecture from random initialization, rather than fine-tuning from ImageNet. This gave a higher validation loss, around 8 years, corresponding to a somewhat worse conditional mean embedding estimate. When we tested on the same 400 samples, the p value dropped to just 0.015, indicating that these tests' Type I error behavior can radically depend on small changes in the quality of the CME estimate.
>
> Now that we have set up the machinery to run these kinds of tests, we will add repeated runs and a more thorough exploration of these comparisons across different test sets, and add the results in the revised version.
>
> ---
>
>
> ## Clarifying what is $q$ when defining $t_n$
>
> Under the null, $n \mathrm{KCI}_n$ converges in distribution to a mixture of chi-squared variables; $q$ is the $1-\alpha$th quantile of that limiting distribution, so that $t_n = q / n$ is the right rejection threshold for $\mathrm{KCI}_n$. We'll clarify this in the text.
>
>
> ---
>
> We sincerely thank the reviewer for the encouraging evaluation and constructive feedback.  We believe the suggested additions will make the paper stronger and more comprehensive.

---

### Official Review · Reviewer_ZYGz · 2025-07-01

**Clarity:** 2
**Significance:** 3
**Originality:** 2
**Rating:** 4
**Confidence:** 3

**Summary:**

This paper provides a comprehensive theoretical analysis of kernel-based conditional independence (KCI) testing. The authors first introduce the KCI framework, and re-interpret it from the perspective of generalized covariance measures (GCM), in a distinct way from [1]. Then, the paper presents an analytical example to illustrate the behavior of the proposed KCI statistic both with and without regression errors. Several simulations were conducted to emphasize the importance of selecting an appropriate kernel of conditioning variables. The theoretical contributions of the paper mainly focus on establishing an error bound of SNR estimator (the KCI statistic divided by its variance), as well as the bounds on the type I error of the test in the presence of regression error. Overall, the authors provide a theoretically sound and carefully argued analysis of KCI, supported by a few simulations that validate their theoretical insights.

[1] Wenjie Wang et al. (2025), Practical Kernel Learning for Kernel-based Conditional Independent Test, Openreview.

**Questions:**

1. I strongly recommend to fix the y-axis when plotting Figure 3 (a), (b). In addition, it is quite surprising that the type II errors of blue and green lines increase with a larger sample size, when $l_C^2 = 10^{-5}$. Moreover, the standard deviation of the type I error becomes noticeably larger with larger sample size. Could the authors provide further explanation for these phenomena?
2. Could you explain where the $\lambda = \ell n^{-1/3}$ in line 874 comes from? This is crucial for the first term in the convergence rate of the SNR. And I noticed Liu et al. also chose a similar $\lambda$ in their Th 6 [2].
3. Could the authors extend their discussion of KCI to also RCIT[3] and SCIT [4]? Given that these methods are highly correlated to GCM and KCI.

[2] Feng Liu et al. (2020), Learning Deep Kernels for Non-Parametric Two-Sample Tests, ICML.

[3] Hao Zhang et al. (2017), Causal Discovery Using Regression-Based Conditional Independence Tests, AAAI.

[4] Hao Zhang et al. (2022), Residual Similarity Based Conditional Independence Test and Its Application in Causal Discovery, AAAI.

**Ethical Concerns:**

["NO or VERY MINOR ethics concerns only"]

**Final Justification:**

The authors finished the rebuttal nicely. They gave extensive extra experiments, tried to explain the abnormal place in their figures, clarified the convergence rate, and I also hope to read their further discussion about RCIT and SCIT. Overall, most of my problems are sloved.

**Limitations:**

yes

**Paper Formatting Concerns:**

There are several issues with the reference formatting, e.g.: ‘NeurIPS.arXiv:1610.02413’ and 'ICML.arXiv:2105.04544'. I strongly advise the authors to carefully revise the reference list. And please check the formula between line 794 and 795.

**Quality:**

3

**Strengths And Weaknesses:**

Strengths:
1. The analyses of the example in Section 4 are detailed and valuable.
2. The work improves understanding of KCI testing, and provides a solid foundation for further theoretical improvements.
3. The authors gave the bound of type one error which is missing from [1].

Weaknesses:
1. It would be beneficial if the authors could provide more extensive simulations to further examine the the proposed CIT and theories. It would be better if the results could be supported by real data.
2. I suggest that the authors consider moving some of the important theoretical results in the appendix into the main text, to better highlight the contributions of the paper. Conversely, certain parts of the main text could be condensed, or even moved to the appendix, such as some part of Section 2, 3.

---

> ### Author Rebuttal · Authors · 2025-07-31
>
> We thank the reviewer for the thoughtful feedback and for recognizing that our work "improves understanding of KCI testing" and "provides a solid foundation for further theoretical improvements." Below, we address the specific comments and outline the improvements we will make based on these suggestions.
>
> ## More Extensive Simulations and Real Data
> We thank the reviewer for this constructive suggestion. We have run substantial additional simulations beyond those in the submitted paper, which we now present to demonstrate how expirical results align with our theory.
>
> To begin with, we have additional simulation results for the settings already considered in the paper; we will discuss a new setting on images afterwards.
>
> ### 1. Setting of Section 4
>
> We compared our power maximization (PM) approach against standard KCI across training sizes 200-1000 (only parts of them are shown due to space limit), using linear kernels on $A$ and $B$, with $\beta=3$ and test size equals to 200. All experiments are repeated 100 times. We report the mean ± standard error.
> The standard KCI use fixed-bandwidth conditioning kernel while PM use learned-bandwidth conditioning kernel:
>
> |Training Size|Type I Error (standard)|Type II Error (standard)|Type I Error (PM)|Type II Error(PM)|
> |-|-|-|-|-|
> |200|0.13±0.034|0.81±0.039|0.10±0.030|0.07±0.026|
> |400|0.11±0.031|0.77±0.042|0.11±0.031|0.01±0.010|
> |600|0.03±0.017|0.67±0.047|0.04±0.020|0.02±0.014|
> |1000|0.03±0.017|0.78±0.041|0.04±0.020|0.00±0.000|
>
>
> Learning the conditioning kernel gives a huge improvement in test power, bringing Type II error from around 75% to almost 0. In this problem, it also maintains reasonable Type I error control, validating our theoretical analysis that the choice of $C$ kernel can be vital to empirical performance. On this problem, the bandwidth optimization stays in a region of reasonable Type I error, as shown in our Figure 2.
>
> ### 2. Complex 3-Dimensional Conditional Dependence Analysis (Setting of Appendix G.3)
>
> In this setting, we have a three-dimensional $C = (C_1, C_2, C_3)$, and use Gaussian kernels on $A$ and $B$, with $\beta=2$. All kernels are Gaussian; power maximization chooses per-dimension bandwidths for the $C$ kernel.
>
> Our analysis looks at two critical scenarios. However, due to space constraints, we present and discuss only one scenario here. This scenario follows the same setting as Appendix G.3 but includes more comprehensive results. The additional results will be provided in the appendix of the revised paper.
>
> #### Scenario: Multi-coordinate dependence
> Here $A$, $B$, and their conditional dependence use different coordinates, e.g. $A$'s marginal distribution depends on $C_1$, $B$'s on $C_2$, and their correlation depends on $C_3$.
>
> With sufficiently trained regressors:
> |Training Size|Type I Error (Standard)|Type II Error (Standard)|Type I Error (PM)|Type II Error (PM)|
> |-|-|-|-|-|
> |200|0.13±0.034|0.22±0.041|0.14±0.035|0.04±0.020|
> |400|0.09±0.029|0.16±0.037|0.12±0.032|0.02±0.014|
> |600|0.07±0.026|0.18±0.038|0.06±0.024|0.00±0.000|
> |800|0.06±0.024|0.14±0.035|0.03±0.017|0.00±0.000|
> |1000|0.03±0.017|0.20±0.040|0.07±0.026|0.00±0.000|
>
> With poorly trained regressors:
> |Training Size|Type I Error (Standard)|Type II Error (Standard)|Type I Error (PM)|Type II Error (PM)|
> |-|-|-|-|-|
> |200|0.49±0.050|0.15±0.036|0.59±0.049|0.09±0.029|
> |400|0.23±0.042|0.20±0.040|0.35±0.048|0.11±0.031|
> |600|0.10±0.030|0.24±0.043|0.15±0.036|0.11±0.031|
> |800|0.06±0.024|0.47±0.050|0.08±0.027|0.20±0.040|
> |1000|0.10±0.030|0.55±0.050|0.10±0.030|0.22±0.041|
>
> The patterns agree well with our explanations that poorly-trained CMEs make Type I error explode compared to well-trained CMEs.
> With insufficiently-trained CMEs, power maximization makes Type I error far worse under this scenario. Though we only focus on one scenario here,  we can also provide additional explanations highlighting the differences between the two scenarios, which will be included in the revised appendix.
>
> We will expand on these and related observations in the final version.
>
>
> ### 3. Real Data Experiments: UTKFace Age Prediction Study
>
> We conducted experiments using the UTKFace dataset, following the methodology from a recent paper:
>
> Y Zhang, Huang, Yang, and Shao. ``Testing Conditional Mean Independence Using Generative Neural Networks.'' *ICML 2025.*
>
> While Zhang et al. did not describe it this way, this test in fact is a KCI test; rather than estimating conditional mean embeddings with ridge regression, however, they use neural networks.
>
> In particular, we (following their setup) took the cropped and aligned UTKFace dataset, and ran a CI test checking whether the age ($A$) is dependent on the whole face image ($B$) when conditioned on the image with the mouth region blacked out ($C$).
>
> In their paper, p-values are above the 5% level when the mouth is covered, suggesting that this region is "not critical for age estimation".  They use ImageNet-pretrained neural networks to perform the regression task, achieving a mean absolute error on validation points of around 6 years. When we reran their code and evaluated on only 400 test samples (single run), the resulting p-value was 0.068 (above the significance level).
>
> We train the same network architecture from random initialization, rather than fine-tuning from ImageNet. This gave a higher validation loss, around 8 years, corresponding to a somewhat worse conditional mean embedding estimate. When we tested on the same 400 samples, the p value dropped to just 0.015, indicating that these tests' Type I error behavior can radically depend on small changes in the quality of the CME estimate.
>
> Now that we have set up the machinery to run these kinds of tests, we will add repeated runs and a more thorough exploration of these comparisons across different test sets, and add the results in the revised version.
>
> ---
>
> ## Clarifying Figure 3 and Paper Organization
>
> The reviewer's attention to detail with respect to this figure is very much appreciated!
>
> With respect to the standard deviations: since these are averages of binary data points (did the test reject or not), the standard *deviation* is actually a function only of the mean ($\sqrt{p (1-p)}$). Here, as the caption says, we plot the standard *error*, which further divides by the square root of the number of runs. Looking into Figure 3, we accidentally ran a total of 500 runs for panel (a) and only 100 for panel (b), explaining the difference; we've increased the number of runs to 500 for all settings to ensure consistency across experiments. Although we cannot present all results here due to space constraints, we will update the results in the revised version.
>
> For the slight increase in Type II error with more training numbers, at very small or very large lengthscales, we don't have a clear explanation for this phenomenon. It might be related to detecting subtle errors in CME estimation in this setting, but it's not clear why that would happen with $\beta \ne 0$ but not with $\beta = 0$, since the distributions given to the CME regression problem are identical.
>
> We also thank the reviewer for the valuable suggestion on the paper organization. We will optimize it to better highlight our contributions.
>
> ---
>
> ## Could you explain where the $\lambda = \ell n^{-1/3}$ in line 874 comes from?
>
> The bound in line 893 works for any choice of $\lambda$; the question, then, is which $\lambda$ to choose to get the best bound. Choosing $\ell n^{-1/3}$ balances between the first $\lambda$ term and the middle two $1 / \sqrt{\lambda n}$ terms, with the $1/\sqrt{n}$ and $1/(n \sqrt\lambda)$ terms not asymptotically relevant. A smaller choice of $\lambda$ would yield a worse rate from the first term, while a larger choice would make the later terms worse. Thus, this choice of $\lambda$ gives the best asymptotic bound.
>
> [6] chose to use simply $\lambda = n^{-1/3}$, concerning themselves only with the asymptotics. Here, we scaled by an arbitrary $\ell$ to emphasize that the theory does not really predict a particular amount of regularization to be optimal (unless we know all the problem-dependent constants, but even then, the constants in the bounds are surely loose enough that this won't be the actually-best choice of regularizaton). The best $\ell$ would be some function of the other problem constants like $\rho$, $s$, $D$, etc.
>
> In fact, the *optimal* choice of $\lambda$ would have slightly different $n$ dependence, to account for the $1/\sqrt n$ and $1 / (n \sqrt\lambda)$ terms. This would be much more complicated, for no asymptotic advantage.
>
> We will add some discussion of this choice, which indeed goes by with absolutely no explanation in the current version.
>
> ---
>
> ## RCIT and SCIT
> We thank the reviewer for suggesting the relationship to RCIT and SCIT.
>
> RCIT is proposed specifically for causal discovery, under the assumption that the data-generating procedure follows additive noise models. Using this assumption, it relaxes the test of $A ⊥ B \mid C$ to simpler unconditional independence tests of $A − f(C) ⊥ B − g(C)$, and $A − f(C) ⊥ C$ or $B − g(C) ⊥ C$.  Since RCIT uses GCM-type statistics on regression residuals $A - f(C)$ and $B - g(C)$, our analysis applies broadly to it.
>
> SCIT, at first glance, seems different enough that it might be difficult to frame in our analysis. It seems more related to the permutation-type tests discussed in Appendix C, which while closely related are hard to directly analyze in the same way in our framework. We'll think more about it.
>
> We will add a short discussion in the revised manuscript to highlight this connection.
>
>
> ---
> We thank the reviewer for the thorough comments, which have led to real improvements in the paper in formal content, experiment, and explanation. Hopefully the reviewer agrees that this has improved our "theoretically sound and carefully argued analysis of KCI" and will consider raising the numerical score for our submission.

---

> > ### Comment · Reviewer_ZYGz · 2025-08-03
> >
> > I sincerely thank the authors for the additional simulations and experiments on the UTKFace dataset, and for the further clarifications they provided in the rebuttal. I would like to keep my rating.

---

### Official Review · Reviewer_397d · 2025-07-02

**Clarity:** 3
**Significance:** 3
**Originality:** 3
**Rating:** 5
**Confidence:** 3

**Summary:**

Aiming to improve the understanding and failure modes of conditional independence (CI) tests, this work provides several novel insights into the hardness of CI testing in practice.
 The authors mainly analyse the Kernel-based conditional independence test (KCI), noting that results also relate closely to the (weighted) Generalized Covariance Measure (GCM) under their formulation of the test in Thm. 2.2. The main insight is that the estimation of conditional mean embeddings presents the main practical limitation of KCI. Specifically, the well-known hardness result by Shah and Peters (2020) does not hold for KCI when using true mean embeddings (Prop. 3.1) but its practical issues arise from the finite-sample estimates of the embeddings. The authors show that this can be somewhat alleviated through the kernel choice (Section 4) but also acknowledge that this comes at the expense of inflating Type I errors, i.e., false positive rejections of independence (Figure 2).

**Questions:**

Minor suggestions and questions include

- If space allows, the density of the text could be reduced. For example, an explanation of Fig. 1 (besides ln. 271) seems to be missing in the text, and some paragraphs could be expanded such as lines 327-335 to explain the insights given in Appendix G.2, G.3.
- a clearer separation of previous insights compared to those that are novel contributions could be helpful, e.g. some of the existing works could be postponed to the Appendix or a related work section.

**Ethical Concerns:**

["NO or VERY MINOR ethics concerns only"]

**Final Justification:**

It seems that all concerns and questions raised during the rebuttal were resolved, such as

- The initial concerns about the novelty of Thm. 2.2 compared to existing results by Daudin 1980 raised by Reviewer v1Tn. This was resolved after additional clarifications by the authors, (some of) which they promise to include in the revised version.
- The arguably small empirical evaluation. In the rebuttal, the authors provided more extensive experiments on synthetic data as well as on real-world data. This strengthens the empirical contribution besides the already strong theoretical one.

All points raised by the reviewers can be addressed without substantial changes of the submitted manuscript.
Overall, I maintain a positive impression of this work and raise the numerical score.

**Limitations:**

No major unadressed limitations or societal impacts.

**Paper Formatting Concerns:**

No major concerns.

**Quality:**

4

**Strengths And Weaknesses:**

The work contributes several novel insights that are likely of interest to the literature. This includes the reframing of KCI to more closely relate it to GCM (Thm. 2); the insight on practical failures of KCI being related to conditional mean embedding estimation which, according to the authors, is not yet acknowledged in existing work; the discussion of appropriate kernel choices (Section 4); and the exploration and cautionary remarks regarding Type I errors (Section 5). Especially in light of the importance of CI testing and widespread use of KCI, this makes this work of solid significance, even though the practical contribution is perhaps limited.
There are no apparent issues regarding quality, all claims have thorough justifications. While some results are inspired from previous insights, such as the strategy motivated in Prop. 4.1. following related settings, the contributions can all in all judged to be of sufficient originality.
Regarding clarity, the writing is dense  but overall well structured around the main claims, and all insights are placed in the context of previous work.

 Overall, this work advances the current understanding of the behavior of KCI and can be judged an important contribution.

---

> ### Author Rebuttal · Authors · 2025-07-31
>
> We thank the reviewer for the detailed and positive assessment of our paper. We greatly appreciate the reviewer for recognizing the significance of our insights regarding conditional mean embeddings and their role in the practical failures of KCI. Below, we address the minor suggestions and clarify how we will revise the manuscript accordingly.
>
> ## Explanation of Figure 1 and Dense Text
> We fully agree with these suggestions for improvement. In the revised version, we will expand the explanation of figures and tables to clearly highlight the intuition behind the experiment and its implications. Additionally, we will slightly reduce text density and restructure some parts to make the narrative more accessible without sacrificing rigor.
>
> We also agree that the insights from Appendices G.2 and G.3 are presented too briefly in the main text. In the revision, we will expand their explanation and integrate a clearer discussion to highlight the key findings and their implications more effectively.
>
>
> ## Distinguish Contributions
>
> On reflection, we also agree that our contributions are not always as distinct as they should be from summaries of prior work. We will revise to address this, including moving some of the discussion of prior work that is currently mixed in Sections 2 and 3 to integrate with the discussion in Appendix C, and further emphasizing in text which results are novel.
>
> ---
> We thank the reviewer for recognizing our work as providing "novel insights," "solid theoretical justification," and "important contribution." We have carefully addressed all minor suggestions and will implement the revisions as described above. We believe that with these revisions, the clarity and accessibility of this paper are further improved, although we can't show the exact revisions in the review process here. Hopefully the reviewer agrees that these minor changes are all that is needed for this "important contribution" and can consider raising the numerical score.

---

> > ### Comment · Reviewer_397d · 2025-08-04
> >
> > Thank you for your response and the proposed modifications regarding clarity. I have no further questions and maintain a positive assessment of this work.

---

### Official Review · Reviewer_v1Tn · 2025-07-03

**Clarity:** 3
**Significance:** 3
**Originality:** 3
**Rating:** 5
**Confidence:** 4

**Summary:**

This paper examines the practical hardness of conditional independence testing with kernels. Specifically, Shah and Peters give a result that conditional independence testing is statistically hard, however their constructive proof gives little insight as to why common kernel based tests fail. In this work, the authors examine the failure of these tests, demonstrating that if we had exact knowledge of the conditional mean embedding the testing problem becomes feasible, constructing a test that is consistent against any fixed alternative. They then study how the choice of kernel impacts the estimation of the conditional mean embedding, due to its centrality to the testing problem.

**Questions:**

- Is Theorem 2.2 he same as condition i) in Lemma 2.2 of [1]?

- Can the authors comment on similarity to goodness of fit tests?

- What would the authors feel are the key limitations of the paper?

**Ethical Concerns:**

["NO or VERY MINOR ethics concerns only"]

**Final Justification:**

Following discussion with the reviewers I now see their paper as a strong contribution to the conditional independence testing literature and so I am changing my score to accept.

**Limitations:**

The limitations are not discussed despite the fact that this box is checked in the checklist.

**Paper Formatting Concerns:**

As mentioned above, the limitations are put as discussed in the checklist but I can find no discussion of them in the paper.

**Quality:**

3

**Strengths And Weaknesses:**

Strengths:
- I find the work to be interesting and novel, it plugs in my opinion a needed gap between the hardness result of Shah and Peters and practical conditional independence testing.

- The observation that access to the true conditional mean embedding leads to a consistent test is a good one and is in line with the cited model X work.

Weaknesses:

- Theorem 2.2 is not new or an extension of Daudin 1980. In fact (without the weighting term) this is the same as condition i) in Lemma 2.2 of [1]. Seeing this classic result described as novel leads me to be concerned about the validity of the rest of the work and the authors familiarity with the area.

-  Whilst I do agree that the observation that the true CME are enough for a valid test is an interesting one, it does seem to me that this result is an example of (or very closely related to) the idea of goodness of fit tests with kernels [2]. Specifically, having access to both $\mu_{A|C}$ and $\mu_{B|C}$ gives us access to $\mu_{A,B|C}$ under the null distribution. Given this, conditional independence testing could be framed as a goodness of fit test for if samples $(a_i,b_i,c_i)$ are consistent with $\mu_{A,B|C}$. In this light a consistent test feels unsurprising due to results of consistency of goodness of fit tests.

Overall whilst this paper seems interesting and is a valid contribution to the literature, my current feeling is that it needs a bit more work before publication as evidenced by the claims of novelty of theorem 2.2 and the incomplete limitations section. Having said this if the authors provide strong arguments I would consider changing my score.

Small note: I would consider changing the phrase "It has a reputation, however, of doing a poor job at controlling Type I error (that is, it falsely identifies conditional dependence too often)." and ideally fine a citation or include an experimental result justifying this as without a strong familiarity with the area this statement is hard to verify.

[1] - Kernel-based Conditional Independence Test and Application in Causal Discovery. Kun Zhang, Jonas Peters, Dominik Janzing, Bernhard Schoelkopf.
[2] - A Kernel Test of Goodness of Fit. A Kernel Test of Goodness of Fit.

---

> ### Author Rebuttal · Authors · 2025-07-31
>
> We sincerely thank the reviewer for recognizing our work to be "novel," "interesting" and a "valid contribution to the literature." We greatly appreciate the constructive feedback provided and will address each question and suggestion in detail below.
>
> ## Clarification on the Novelty of Theorem 2.2:
>
> We'd like to clarify that our Theorem 2.2 is in fact meaningfully different from the results of Daudin, which were repeated as Lemma 2.2 of Zhang et al. The difference is a little obfuscated by the notational decisions of Daudin that were also used by Zhang et al., in not explicitly writing out the arguments to the functions in question. Let's restate the results in the same terms, to compare.
>
> Theorem 2.2 of our paper:
> > Random variables $A$ and $B$ are conditionally independent given $C$ if and only if
> >
> > $\mathbb E_C\Bigl[ w(C) \; \mathbb E\bigl[ (f(A) - \mathbb E[ f(A \mid C)]) \, (g(B) - \mathbb E[ g(B \mid C)]) \bigr] \Bigr] = 0$
> >
> > for all square-integrable functions $f \in L^2_A$, $g \in L^2_B$, and $w \in L^2_C$.
>
> Compare with Lemma 2.2 of Zhang et al., which stated results of Daudin, rewritten in our notation:
> > The following conditions are equivalent to one another:
> >
> > (i) $A$ is conditionally independent of $B$ given $C$.
> >
> > (ii) $\mathbb E\Bigl[ \tilde f(A, C) \tilde g(B, C) \Bigr] = 0$ for all $\tilde f \in L^2_{AC}$ with $\mathbb E[ \tilde f(A, C) \mid C] = 0$ and $\tilde g \in L^2_{BC}$ with $\mathbb E[ \tilde g(B, C) \mid C] = 0$.
> >
> > (iii) $\mathbb E\Bigl[ \tilde f(A, C) g(B, C) \Bigr] = 0$ for all $\tilde f \in L^2_{AC}$ with $\mathbb E[ \tilde f(A, C) \mid C] = 0$ and $g \in L^2_{BC}$.
> >
> > (iv) $\mathbb E\Bigl[ \tilde f(A, C) \bigl( g'(B) - \mathbb E[g'(B) \mid C] \bigr) \Bigr] = 0$ for all $\tilde f \in L^2_{AC}$ with $\mathbb E[ \tilde f(A, C) \mid C] = 0$ and $g' \in L^2_{B}$.
> >
> > (v) $\mathbb E\Bigl[ \tilde f(A, C) \, g'(B) \Bigr] = 0$ for all $\tilde f \in L^2_{AC}$ with $\mathbb E[ \tilde f(A, C) \mid C] = 0$ and $g' \in L^2_{B}$.
>
> The key difference is that all versions of Daudin's result depend on a centered function $\tilde f$ of _both_ $A$ and $C$.
>
> Zhang et al. construct the KCI statistic through centering in an RKHS on $(A, C)$, which they denote $\ddot{X}$. They use a product kernel on $(A, C)$; the recent papers of Pogodin et al. (2023, 2024) noticed that this allows separating out the $C$ dependence, as done in the statements of "the KCI operator" in those papers. (This also helped substantially in avoiding the need for regressing the identity function on $C$, an ill-posed regression problem in rich kernel spaces that can actually cause serious problems; see discussion in Appendix B.1 of Pogodin et al. 2024.) Wang et al. (2025) did the same split decomposition of KCI in their equation (3), yielding the same formula.
>
> While we end up with the same result, the advantage of Theorem 2.2 is that it allows a far more direct path to justifying the formula, rather than the roundabout method through product kernels. The role of the $C$ kernel as roughly "selecting" $C$ values for consideration, we think, is much easier to see in the Daudin-style characterization with explicit test functions, rather than relying on a connection to L2-universal product kernels.
>
> This result also helps provide a much more direct connection to, and direct justification for, weighted GCM. In particular, it makes it easier to consider distribution-level properties and finite-sample estimators separately for GCM and weighted GCM than the approach used by Scheidegger et al; our Theorem 2.2 makes clear that the "distribution-level" weighted GCM with sufficiently rich regression and weighting functions can characterize conditional independence.
>
> We will add further clarification of these points in the revised version; thank you for raising the concern.
>
> [i] Pogodin, Roman, et al. "Efficient Conditionally Invariant Representation Learning." *ICLR (2023).*
>
> [ii] Pogodin, Roman, et al. "Practical kernel tests of conditional independence." *arXiv preprint arXiv:2402.13196 (2024).*
>
> [iii] Wang, Wenjie, et al. "Practical Kernel Learning for Kernel-based Conditional Independent Test." *(2025).*
>
> ---
>
> ## Clarification on Relation to Goodness-of-Fit Testing
>
> We appreciate the connection between our test and kernel-based goodness-of-fit testing [2]. Below, we briefly clarify why our construction and the test discussed in [2] differ conceptually and practically:
>
> - The construction described by the reviewer indeed starts from conditional mean embeddings (CMEs) $\mu_{A|C}$ and $\mu_{B|C}$, then attempts to use them directly to obtain $\mu_{A,B|C}$ under the null. However, note that this would also require explicit knowledge of $\mu_{C}$, to reconstruct the joint embedding $\mu_{A,B,C}$. Such explicit knowledge (or estimation) of $\mu_C$ goes slightly beyond the assumptions typically made for conditional independence testing.
>
>
> - The goodness of fit test of [2] is designed specifically for continuous distributions with a known score function or gradient information. In contrast, in our conditional independence setting, no explicit model providing gradient information is available. The KSD-based tests from [2], then, don't really make sense here. (The tests of [2] also provide only pointwise asymptotic validity, but this problem is fixable.)
>
> - Alternatively, one might consider an MMD-based goodness of fit test, comparing the estimated embedding $\hat\mu_{A,B,C}$ to the known embedding $\mu_{A,B,C}$ (assuming, as before, that we also know $\mu_C$, or by estimating it). Indeed, such an MMD-based approach combined with a permutation test might offer finite-sample level control, and this could be another way to test conditional independence in the setting where we know $\mu_C$, $\mu_{A\mid C}$, and $\mu_{B \mid C}$.
>
>
> We intentionally presented Proposition 3.1 as a straightforward consequence of Hoeffding’s inequality for U-statistics (hence labeling it a Proposition rather than a Theorem), not as a surprising or controversial result. Rather, its purpose is precisely to highlight clearly the importance of accurately estimating the conditional mean embeddings and illustrate explicitly why inaccuracies in their estimation degrade test performance.
>
> In short, we fully agree that the reviewer’s suggested viewpoint casting conditional independence testing as a goodness of fit test is an insightful alternative interpretation – but this interpretation complements, rather than undermines, our main point about the central role of accurate CME estimation. We will add a discussion of these connections in the revised manuscript.
>
> ---
>
> ## Clarification on Limitations
>
> We agree with the reviewer that the limitations section deserves explicit expansion, and we will do so in our revised manuscript. In particular, we see our main limitation as the one identified in the last sentence of the main body of the submission: while we have clearly identified and rigorously analyzed a core issue underlying KCI-like methods (the crucial role of accurately estimating conditional mean embeddings), we have not yet provided a definitive solution to fully overcome this challenge.
>
> Additionally, our analysis does not cover all existing conditional independence tests. Although we demonstrated that our analysis broadly applies to the class of KCI-type tests (explicitly including methods similar to the Generalized Covariance Measure, GCM), there remain popular CI tests that appear outside this framework -- for instance, the RBPT test discussed in Appendix C. While we think the underlying ideas should also apply to RBPT-type tests, given the different structure, doing so would require further analysis.
>
> We will make these limitations explicit and elaborate clearly upon them in the revised manuscript. We thank the reviewer for highlighting this valuable aspect for improvement.
>
> ---
>
> ## Clarification on the Type I Error Statement
> We appreciate the reviewer’s suggestion regarding the statement: *"It has a reputation, however, of doing a poor job at controlling Type I error (that is, it falsely identifies conditional dependence too often)."* Without reference to clear empirical evidence or citations, this claim does appear unsubstantiated to the reader.
>
> In the revised manuscript, we will explicitly support this claim by citing relevant recent studies, including Shah and Peters (2019) and Pogodin et al. (2024). Additionally, we provide empirical evidence in Appendix G.3 Table 1 (see also our rebuttal of empirical results to reviewer ZYGz), which demonstrates concrete scenarios in which KCI fails to properly control Type I error at nominal significance levels. We will revise to make the justification for this statement clearer.
>
> [iv] Shah, Rajen D., and Jonas Peters. "The hardness of conditional independence testing and the generalised covariance measure." *(2019)*
>
> [ii] Pogodin, Roman, et al. "Practical kernel tests of conditional independence." *arXiv preprint arXiv:2402.13196 (2024).*
>
> ---
>
> We have made every effort to address all questions raised and believe that the revisions respond to the reviewer’s concerns. If the reviewer feels that these clarifications resolve the issues, we would be grateful if the reviewer might consider raising the score. If there are any remaining uncertainties, we would be glad to provide further explanations or discussion to resolve them.

---

> > ### Author Response · Authors · 2025-08-07
> >
> > Hi — thanks again for your review of our paper. Since the discussion period is moving towards a close, we wanted to check in whether we’ve addressed your concerns: in particular,
> > - the form of Theorem 2.2 _is_ a novel extension of prior results, and we will make the differences from Daudin’s version clearer in the text;
> > - we appreciate the connection to goodness-of-fit testing and will mention this as well, but do not think this especially affects the story of our paper;
> > - we will expand the explicit discussion of the limitations at the end of the paper.
> >
> > If you have any remaining concerns or further questions, please let us know!

---

> > > ### Comment · Reviewer_v1Tn · 2025-08-07
> > >
> > > I have read the reviewers response, am satisfied with their rebuttal and will increase my score accordingly. Specifically I am satisfied with the novelty of their theorem and the difference between their approach and goodness of fit testing. I thank the authors for their thorough and detailed response.

---

### Decision · Program_Chairs · 2025-09-17

**Decision:**

Accept (spotlight)

**Comment:**

This paper investigates the Kernel-based Conditional Independence (KCI) test and points out where the “practical hardness” of KCI comes from: the estimation of conditional mean embeddings. With oracle embeddings, a threshold yields a finite-sample valid, consistent test. The paper next analyzes kernel choice on C and shows that KCI’s power is exquisitely sensitive to the C-kernel bandwidth. Then the paper proposes selecting the C-kernel by maximizing an asymptotic signal-to-noise ratio criterion. Finally, the paper discusses the Type-I Error Inflation in the presence of regression errors.

Strengths：All reviewers agree that the paper contributes several novel insights and understandings, especially plugging the gap between the negative result of Shah and Peters and practical conditional independence testing. Reviewer 397d further emphasizes that the paper relates KCI more closely to GCM. Reviewer web4 appreciates that the paper presents a detailed background for unfamiliar readers.

Weaknesses：
The paper would benefit from diverse real-world datasets. Reviewer 397d and ZYGz mentioned that some parts of the main text can be improved.

The most important reasons for the decision：

I appreciate the novelty and insight this paper provided, including but not limited to:

1) Identify the major factors underlying KCI’s practical behavior.

2) The connection between KCI and GCM.

3) Help readers advance understanding of why CI tests fail in practice.

4) The quantitative bounds pinpoint regression error as the bottleneck.

The results will likely influence how CI is used in causal discovery and other related areas.

Summarization of the discussion and changes during the rebuttal period:
Reviewer v1Tn raises the problem on the relation between Theorem 2.2 and Daudin’s work. After the rebuttal phase, the reviewer agrees with the novelty of their theorem.

Overall comments:

After the rebuttal phase, the paper received three accept and one borderline accept, resulting in unanimous acceptance. CI testing is a very fundamental area. This paper turns an impossibility result in CI testing into practical guidance. Rather than re-stating the Shah and Peters’s hardness theorem, the paper pinpoints where things go wrong in practice: errors in estimating conditional mean embeddings, which reframes “CI is impossible” into “CI is doable if you control specific estimation errors.” This core insight, together with the depth of the contribution, makes the work stand out. All reviewers agree that the proposed methods are both theoretically novelty and interesting. Overall, the quality of the work is high and solid, and it is likely to be of interest to the community. Therefore, I recommends acceptance as a spotlight.